# GenCape: Structure-Inductive Generative Modeling for Category-Agnostic Pose Estimation

**Jiyong Rao[1], Yu Wang[1]\* and Shengjie Zhao[1]\***
[1] School of Computer Science and Technology, Tongji University

## Abstract

Category-agnostic pose estimation (CAPE) aims to localize keypoints on query images from arbitrary categories, using only a few annotated support examples for guidance. Recent approaches either treat keypoints as isolated entities or rely on manually defined skeleton priors, which are costly to annotate and inherently inflexible across diverse categories. Such oversimplification limits the model's capacity to capture instance-wise structural cues critical for accurate pixel-level localization. To overcome these limitations, we propose **GenCape**, a **Gen**erative-based framework for **CAPE** that infers keypoint relationships solely from image-based support inputs, without additional textual descriptions or predefined skeletons. Our framework consists of two principal components: an iterative Structure-aware Variational Autoencoder (i-SVAE) and a Compositional Graph Transfer (CGT) module. The former infers soft, instance-specific adjacency matrices from support features through variational inference, embedded layer-wise into the Graph Transformer Decoder for progressive structural priors refinement. The latter adaptively aggregates multiple latent graphs into a query-aware structure via Bayesian fusion and attention-based reweighting, enhancing resilience to visual uncertainty and support-induced bias. This structure-aware design facilitates effective message propagation among keypoints and promotes semantic alignment across object categories with diverse keypoint topologies. Experimental results on the MP-100 dataset show that our method achieves substantial gains over graph-support baselines under both 1- and 5-shot settings, while maintaining competitive performance against text-support counterparts.

## 1 Introduction

Category-Agnostic Pose Estimation (CAPE) Xu et al. (2022a); Shi et al. (2023); Hirschorn & Avidan (2024); Rusanovsky et al. (2025); Ren et al. (2024); Chen et al. (2025a) has emerged as a fundamental yet challenging task in computer vision, aiming to localize semantic keypoints on arbitrary object categories using only a handful of annotated support samples. Unlike conventional 2D pose estimation task Sun et al. (2019); Xu et al. (2022b); Yuan et al. (2021); Rao et al. (2025), which depends heavily on predefined templates or class-specific priors, CAPE requires robust generalization across semantically diverse and structurally heterogeneous object classes. This capability extends the applicability of pose estimation from closed-world scenarios to open-world scenarios, enabling scalable deployment in domains such as human motion analysis Zheng et al. (2023); Yang et al. (2023), cross-species behavior understanding Ye et al. (2024); Stoffl et al. (2024), and robotic manipulation Zheng et al. (2025); Ma et al. (2024) in dynamic environments.

In the CAPE paradigm, the objective is to estimate keypoints for objects from novel categories, conditioned on a few annotated support images. Despite recent advancements, existing CAPE approaches are hindered by two critical limitations. On the one hand, many existing approaches either treat keypoints as isolated semantic entities, neglecting the latent spatial dependencies essential for accurate pixel-level localization, or rely heavily on external priors such as manually pre-defined skeleton connections or auxiliary textual descriptions. These external priors not only incur high

---

\*Corresponding author: `yuwangtj@yeah.net` and `shengjiezhao@tongji.edu.cn`

annotation overhead but also restrict the model's adaptability to novel instances with large pose variations, non-rigid deformation, or diverse structural characteristics.

On the other hand, the stochastic nature of support sets selection in a few-shot setting makes CAPE methods particularly vulnerable to the low quality support examples. In real-world scenarios, support images may contain severe occlusions, incomplete annotations, or structural discrepancies relative to the query, which can misguide structural inference and significantly impair prediction accuracy and generalization. Together, these limitations underscore the need for a more flexible, data-driven approach to structure modeling and robust support-query adaptation.

To address the limitations of fixed priors and structural rigidity in CAPE, we propose **GenCape**, a generative latent structure learning framework that automatically infers keypoint relationships, represented as latent adjacency matrices, exclusively from image-based support inputs, without any external priors, as illustrated in Figure 1. At its core of, GenCape integrates two complementary components: an *iterative Structure-aware Variational Autoencoder (i-SVAE)* and a *Compositional Graph Transfer (CGT)* module. Specifically, the i-SVAE leverages variational inference to learn a distribution over instance-specific graph structures, iteratively generating and refining latent adjacency matrices that serve as flexible and data-driven structural priors. Compared to the recently related deterministic approach SDPNet Ren et al. (2024), this generative formulation captures the epistemic uncertainty in sparse and ambiguous support signals, allowing for more expressive and robust message passing within the Graph Transformer Decoder. The progressive refinement enables the model to propagate contextual cues across spatially correlated keypoints and adapt to complex object configurations. In parallel, the CGT module dynamically aggregates multiple latent graph hypotheses into a query-conditioned representation through a principled Bayesian fusion and an attention-based re-weighting strategy. This dynamic compositional mechanism explicitly accounts for support-query inconsistencies and mitigates the adverse impact of noisy or misleading support examples caused by occlusion, deformation, or pose variation. Together, these modules enhance structural generalization and resilience to support noise, setting a new direction for few-shot keypoint reasoning under the CAPE paradigm. Remarkably, our approach surpasses the performance of existing CAPE methods, showcasing a new state-of-the-art performance.

In summary, our contributions are as follows:

- We introduce **GenCape**, a novel generative framework for category-agnostic pose estimation, which incorporates an *iterative Structure-aware Variational Autoencoder (i-SVAE)* to infer latent, instance-specific skeletons solely from image-based support sets, eliminating the need for predefined anatomical priors or textual descriptions.

- We propose a *Compositional Graph Transfer (CGT)* mechanism that dynamically aggregates multiple structural hypotheses into a unified, query-conditioned graph through attention-guided fusion, significantly enhancing robustness under ambiguous, noisy, or structurally mismatched support scenarios.

- Our framework achieves new state-of-the-art results on the representative and challenging **MP-100** benchmark under both 1-shot and 5-shot settings, surpassing existing methods by a substantial margin of **+1.59%** mPCK averaged across evaluation thresholds, without relying on any external structural or textual annotations.

## 2 RELATED WORK

### 2.1 CATEGORY-AGNOSTIC POSE ESTIMATION

Category-agnostic pose estimation (CAPE) Xu et al. (2022a) has emerged as a compelling generalization of conventional pose estimation Sun et al. (2019); Yu et al. (2021); Xu et al. (2022b; 2025); Rao et al. (2025) by localizing keypoints for arbitrary category objects with only a few annotated support images. The pioneering POMNet Xu et al. (2022a) employed a metric-learning paradigm to match support and query features in latent space, while CapeFormer Shi et al. (2023), a two-stage refinement framework that first produces initial keypoint proposals and then refines their positions in a second stage. Recent efforts Hirschorn & Avidan (2024); Liang et al. (2024) further extended this paradigm by integrating fixed skeleton priors into graph reasoning modules to capture latent keypoint relations. However, it remains limited by the rigidity of manually defined structures, which

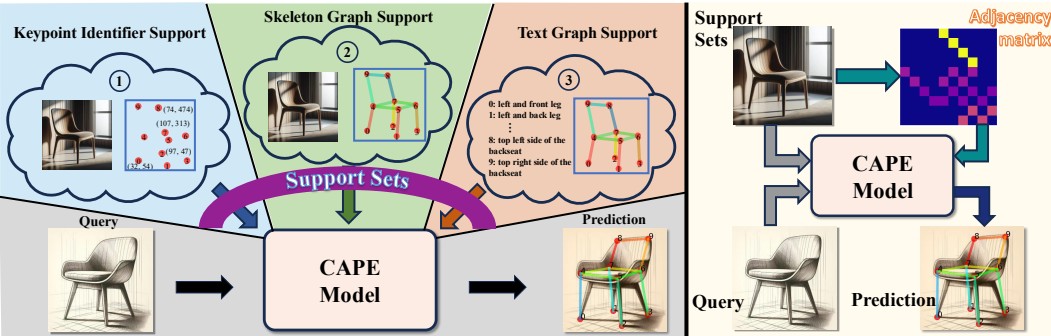

Figure 1: **Left:** Existing CAPE frameworks rely on additional structured priors within the support set, such as ① keypoint identifiers, ② fixed skeleton graphs, or ③ textual descriptions with skeleton graphs, to enhance structural reasoning. **Right:** In contrast, our framework directly infers latent keypoint relationships solely from support images, learning instance-specific *adjacency matrices* without relying on handcrafted priors.

constrains adaptability to novel categories with topological variation. WeakShot Chen et al. (2025b) learns category-agnostic keypoints via diffusion-based keyness prediction and correspondence transfer. Another line of works Yang et al. (2024); Kim et al. (2024); Lu et al. (2024); Rusanovsky et al. (2025) also leverage textual descriptions for guidance, enhancing category-agnostic generalization but still depending on auxiliary language priors. Most relevant to our work, SDPNet Ren et al. (2024) adopts a discriminative approach by predicting a fixed adjacency matrix from support features. However, it lacks mechanisms to model structural uncertainty, limiting robustness under support-query mismatch. In contrast, we propose a generative framework that infers flexible, instance-specific graphs from support images, enabling more adaptive and resilient structure modeling.

## 2.2 LATENT STRUCTURE LEARNING FOR POSE ESTIMATION

Latent structure learning has been widely adopted to reason inter-keypoint dependencies in pose estimation. Early approaches Wang et al. (2020); Hassan & Hamza (2023) leverage graph convolutional networks to refine keypoint predictions by modeling predefined skeletal connections, which restrict applicability to human- or hand-specific topologies. Generative models, particularly variational frameworks like the Variational Graph Autoencoder (VGAE) Kipf & Welling (2016) and CVAM-Pose Zhao et al. (2024) have demonstrated promise in capturing structural variability and uncertainty, but typically remain tied to specific classes or predefined topologies. More recently, V-VIPE Levy & Shrivastava (2024) leverages a variational autoencoder framework to learn a view-invariant latent pose representation. ProPose Han et al. (2025) reformulates 3D human pose estimation as a probabilistic generative task by modeling instance-level pose distributions, enabling uncertainty-aware and sample-efficient inference. Following these advancements, we explore a generative formulation to learn instance-level latent structures, aiming to enhance generalization and move beyond the reliance on predefined priors. While learning latent structures improves flexibility, it remains insufficient under the CAPE setting, where support sets are sampled stochastically. This insight suggests a mechanism that aggregates multiple latent graphs into a query-conditioned structure, dynamically emphasizing support information most aligned with the query.

## 3 METHOD

To effectively learn optimal keypoint structural dependencies and eliminate the adverse effects of inappropriate support sets, we propose a generative-based framework tailored for Category-Agnostic Pose Estimation (CAPE). This method is grounded in GraphCape Hirschorn & Avidan (2024) framework without reliance on skeleton priors. We begin by presenting a concise overview of the pipeline before introducing our generative graph learning module and query-aware fusion technique.

### 3.1 OVERALL PIPELINE

The goal of CAPE is to estimate the locations of semantic keypoints $\hat{\mathbf{K}}_q \in \mathbb{R}^{M_c \times 2}$ for a query image $\mathbf{I}_q$, given a small set of annotated exemplars from an unseen category, where $M_c$ denotes the maximum possible number of keypoints. In the $N$-shot setting, we are provided with a set of

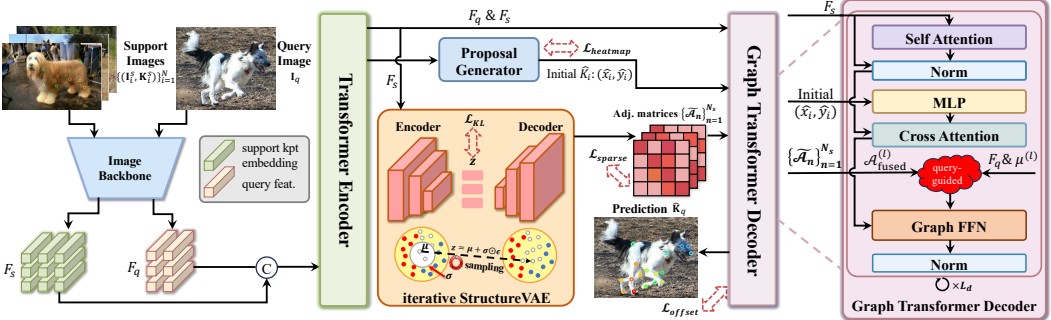

Figure 2: **Architecture Overview.** Our approach utilizes a pre-trained backbone to extract image features, which are refined by a transformer encoder through self-attention. A proposal generator is employed alongside a graph transformer decoder. Subsequently, we employ iterative StructureVAE to generate probabilistic adjacency matrices, and integrates them to a query-aware graph in the graph transformer decoder, improving localization accuracy by graph-oriented decoding.

$N$ support pairs $\{(\mathbf{I}_i^s, \mathbf{K}_i^s)\}_{i=1}^N$, where each support image $\mathbf{I}_i^s$ is annotated with a set of keypoints $\mathbf{K}_i^s \in \mathbb{R}^{M_c \times 2}$ for category $c$ (which may vary in keypoint count $M_c$). Initially, a shared backbone $\phi(\cdot)$ extracts visual features $F_q = \phi(\mathbf{I}_q)$ and $F_s' = \phi(\mathbf{I}^s)$. The support features are then aggregated with their corresponding keypoint targets to produce keypoint-aware embeddings $F_s \in \mathbb{R}^{M \times D}$, where $M$ represents the maximum number of potential keypoints, and $D$ is the embedding size. Then, a similarity-aware proposal generator computes correlations between $F_s$ and $F_q$, yielding position proposals $P \in \mathbb{R}^{M \times 2}$. As shown in Figure 2, to model inter-keypoint dependencies and learn flexible skeleton knowledge, we introduce an *iterative Structure-aware Variational Autoencoder (i-SVAE)* that infers a latent adjacency matrix $\mathcal{A} \in \mathbb{R}^{M \times M}$ conditioned on $F_s$. This probabilistic graph captures instance-specific keypoint relations and is fused with visual cues in the graph transformer decoder. We further mitigate visual uncertainty and reduce the adverse effects of improper support images through a *Compositional Graph Transfer (CGT)* strategy, which aggregates multiple latent hypotheses into a query-aware graph. This composition is injected into the graph transformer decoder to guide self- and cross-attention, progressively refining keypoint predictions.

## 3.2 ITERATIVE STRUCTURE LEARNING

In CAPE, the support and query mismatch in visibility, poses and topologies, making structural alignment essential. Our framework addresses this discrepancy jointly with *i-SVAE* that learns layer-wise, instance-specific graphs from support features,, and *CGT* that adapts them to the query. Most CAPE methods make oversimplified assumptions: either modeling keypoints as independent entities or relying on manually defined priors. Such assumptions hinder the capture of topological consistency and pose variability across instances. To this end, we reformulate structure inference as a graph learning problem: keypoints are nodes, and their relationships are encoded in a latent adjacency matrix. Instead of using static, category-specific graphs, we propose an *i-SVAE* that learns and refines instance-specific keypoint graphs across decoding stages.

**Graph Formulation.** Let the graph $\mathcal{G} = (\mathcal{V}, \mathcal{E})$ consist of $M$ keypoint nodes $v_i \in \mathcal{V}$ with initial node feature matrix $\mathcal{X} \in \mathbb{R}^{M \times D}$, and an initial noisy adjacency matrix $\mathcal{A}^{(0)} \in \mathbb{R}^{M \times M}$ encoding edge set $\mathcal{E}$. Our goal is to learn a function $f : \mathcal{F}_s \mapsto \mathcal{A}$ that maps support keypoint embeddings $F_s$ to a soft adjacency matrix $\mathcal{A}$, capturing latent inter-keypoint dependencies. We formulate this within the principled Variational Graph Autoencoder (VGAE) Kipf & Welling (2016), wherein the graph structure is treated as a latent variable: $q_\phi(\mathbf{z}|F_s)$, $\mathbf{z} \in \mathbb{R}^{D_z}$, with $q_\phi$ denoting the approximate posterior and $\mathbf{z}$ the latent code. The code is decoded into a soft adjacency matrix that models probabilistic keypoint connectivity.

**Iterative StructureVAE.** The i-SVAE consists of two components: a probabilistic encoder that parameterizes a latent graph distribution, and a decoder that constructs the adjacency matrix from this latent space. A detailed figure is shown in the supplementary material. At each layer $l$, given support node embeddings $F_s^{(l)} \in \mathbb{R}^{M \times D}$ from the previous graph transformer decoder layer, we

first perform variational inference to estimate the latent structural embeddings:

$$[\boldsymbol{\mu}^{(l)}, \log(\boldsymbol{\sigma}^{(l)})] = \text{Enc}(F_s^{(l)}), \quad \boldsymbol{\mu}^{(l)}, \log \boldsymbol{\sigma}^{(l)} \in \mathbb{R}^{D_z},$$
$$q_\phi(\mathbf{z}^{(l)} \mid F_s^{(l)}) = \mathcal{N}(\mathbf{z}^{(l)}; \boldsymbol{\mu}^{(l)}, \text{diag}(\boldsymbol{\sigma}^{(l)})), \tag{1}$$

where Enc denotes the graph encoder that produces the approximate posterior distribution over the latent graph codes $\mathbf{z}$. We then employ the reparameterization trick to sample $\mathbf{z}^{(l)} \in \mathbb{R}^{D_z}$:

$$\mathbf{z}^{(l)} = \boldsymbol{\mu}^{(l)} + \boldsymbol{\sigma}^{(l)} \odot \boldsymbol{\epsilon}, \quad \boldsymbol{\epsilon} \sim \mathcal{N}(0, \mathbf{I}), \tag{2}$$

Next, the latent code $\mathbf{z}$ is passed through a fully connected decoder to construct the adjacency matrix $\hat{\mathcal{A}} = \text{Dec}(\mathbf{z}) \in \mathbb{R}^{M \times M}$, where Dec represents the graph decoder. To ensure the undirectionality and interpretability of the adjacency matrix, we symmetrize and normalize it row-wise:

$$\hat{\mathcal{A}}^{(l)}_{\text{sym}} = \frac{1}{2}(\hat{\mathcal{A}}^{(l)} + \hat{\mathcal{A}}^{(l)\top}), \quad \tilde{\mathcal{A}}^{(l)} = \text{norm}(\hat{\mathcal{A}}^{(l)}_{\text{sym}}), \tag{3}$$

Then, we take the *CGT* strategy to fuse multiple sampled adjacency matrices into a unified, query-aware graph, which is introduced in the next subsection. This fusion strategy is performed via Bayesian averaging, reweighted by graph-level uncertainty and relevance to the query features:

$$\tilde{\mathcal{A}}^{(l)}_{\text{final}} = \text{CGT}(\{\tilde{\mathcal{A}}^{(l)}_n\}, \{\boldsymbol{\mu}^{(l)}_n\}, F_q). \tag{4}$$

To ensure meaningful latent representations and regulate structural uncertainty, we minimize the Kullback-Leibler divergence between the approximate posterior $q_\phi(\mathbf{z} \mid \mathbf{X})$ and a Gaussian prior $p(\mathbf{z}) = \mathcal{N}(0, \mathbf{I})$. Besides, we impose an $\ell_2$ sparsity constraint on the learned adjacency to encourage minimal and interpretable connectivity. The total *i-SVAE* loss at the $l$-th decoder layer is defined as:

$$\mathcal{L}^{(l)}_{\text{VAE}} = \mathcal{L}^{(l)}_{KL} + \beta \cdot \mathcal{L}^{(l)}_{\text{sparse}} = \underbrace{D_{\text{KL}}\left[q_\phi(\mathbf{z}^{(1)}|F_s^{(l)}) \parallel p(\mathbf{z}^{(l)})\right]}_{\text{Prior Regularization}} + \underbrace{\frac{\beta}{M^2}\lambda\|\tilde{\mathcal{A}}^{(l)}_{\text{final}}\|^2_F}_{\text{Sparse Matrix}}, \tag{5}$$

where the hyper-parameter $\beta = 0.1$. To leverage the learned structural priors, we follow the Graph-Cape and incorporate a graph convolutional layer conditioned on the final matrix $\tilde{\mathcal{A}}^{(l)}_{\text{final}}$.

$$F_s^{(l+1)} = \sigma\left(W_{\text{adj}}F_s^{(l)}\tilde{\mathcal{A}}^{(l)}_{\text{final}} + W_{\text{self}}F_s^{(l)}\right), \tag{6}$$

where $W_{\text{adj}}, W_{\text{self}} \in \mathbb{R}^{D_{\text{out}} \times D}$ are learnable weights, and $\sigma(\cdot)$ denotes ReLU activation. The first term aggregates features from semantically or spatially connected neighbors, while the second term retains individual node semantics via self-transformation.

final skeleton $\tilde{\mathcal{A}}^{(l)}_{\text{final}}$ serves as the structural guidance for message passing:

$$F_s^{(l+1)} = \text{GCN}(F_s^{(l)}, \tilde{\mathcal{A}}^{(l)}_{\text{final}}), \quad P_q^{(l+1)} = \sigma\left(\sigma^{-1}(P_q^{(l)}) + \text{MLP}(F_s^{(l+1)})\right), \tag{7}$$

where $P_q^{(l)} \in \mathbb{R}^{K \times 2}$ is the predicted keypoint locations at $l$-th layer used for intermediate supervision, with the output from the final layer as the final keypoint prediction. And $\sigma$ and $\sigma^{-1}$ are the sigmoid and its inverse function.

By embedding *i-SVAE* within each decoder layer, our method performs iterative structural refinement, progressively updating latent pose graphs in response to evolving visual semantics and localization cues. This layer-wise iterative design enables the model to capture diverse structural patterns and encode high-order keypoint dependencies, thereby strengthening relational reasoning and improving generalization to novel categories. See Appendix A.1 for further details.

## 3.3 COMPOSITIONAL GRAPH TRANSFER

While *i-SVAE* enables layer-wise modeling of instance-specific pose graphs, its stochastic sampling process introduces uncertainty across multiple latent graphs. To this end, we propose *Compositional Graph Transfer (CGT)*, a query-aware graph fusion mechanism that aggregates multiple sampled adjacency matrices into a robust and expressive structural representation. Specifically, given a set

of $N_s$ latent graphs sampled from *i-SVAE* at the $l$-th decoder layer, denoted as $\{\tilde{\mathcal{A}}_n^{(l)}\}_{n=1}^{N_s}$, each associated with a latent distribution parameterized by $(\boldsymbol{\mu}^{(l)}, \boldsymbol{\sigma}^{(l)})$, our goal is to construct a matrix $\mathcal{A}_{\text{final}} \in \mathbb{R}^{M \times M}$ that best reflects the robust structure relationships among keypoints conditioned on the query context, while simultaneously alleviating over-reliance on the support-driven guidance.

To achieve this, we adopt a Bayesian confidence-weighted aggregation strategy. Firstly, we define the confidence of each sampled graph as the inverse of the total variance:

$$w_n = \frac{1}{\sum_{i=1}^{D_z} \boldsymbol{\sigma}_{n,i}^{(l)} + \epsilon}, \quad \tilde{w}_n = \frac{w_n}{\sum_{m=1}^{N_s} w_m}, \tag{8}$$

where $\epsilon = 1e^{-6}$ is a small constant to ensure numerical stability. These normalized weights $\tilde{w}_n$ reflect the epistemic uncertainty of each latent sample and serve to guide the fusion process. The fused adjacency is then computed via a weighted average: $\tilde{\mathcal{A}}_{\text{fused}}^{(l)} = \sum_{n=1}^{N_s} \tilde{w}_n \cdot \tilde{\mathcal{A}}_n^{(l)}$. To further align the fuse structure with query-specific evidence, we incorporate query-guided gating. Let $F_q \in \mathbb{R}^{hw \times D}$ denote the query feature, where $[h, w]$ denotes the patch size in image backbone. We compute attention-based gating scores $\alpha^{(l)}$ by comparing global query descriptors with each means $\mu^{(l)}$:

$$\alpha^{(l)} = \frac{\text{sim}(\text{Pool}(F_q), \boldsymbol{\mu}^{(l)})}{\sum_{l=1}^{L_d} \text{sim}(\text{Pool}(F_q), \boldsymbol{\mu}^{(l)})}, \tag{9}$$

where $\text{sim}(\cdot, \cdot)$ denotes cosine similarity and Pool is a global average pooling operator. The final fused graph becomes $\tilde{\mathcal{A}}_{\text{final}}^{(l)} = \sum_{l=1}^{L} \alpha^{(l)} \cdot \tilde{\mathcal{A}}_{\text{fused}}^{(l)}$, where $L \in [1, L_d]$ is the current decoder layer. The fusion process enhances robustness against sampling stochasticity and grounds structural reasoning in the visual context of the query. The resulting graph $\tilde{\mathcal{A}}_{\text{final}}^{(l)}$ is propagated into the GCN layer, enabling structure-aware refinement of keypoint predictions. See Appendix A.1 for CGT details.

### 3.4 TRAINING AND INFERENCE

For the category-agnostic pose estimation task, we employ the commonly used loss Shi et al. (2023); Hirschorn & Avidan (2024); Rusanovsky et al. (2025) $\mathcal{L}_{pred}$:

$$\mathcal{L}_{pred} = \lambda_{heatmap} \cdot \mathcal{L}_{heatmap} + \mathcal{L}_{offset},$$

$$\mathcal{L}_{\text{heatmap}} = \frac{1}{M_c \cdot H \cdot W} \sum_{i=1}^{M_c} \left\| \hat{\mathbf{H}}_i - \mathbf{H}_i \right\|, \quad \mathcal{L}_{\text{offset}} = \frac{1}{L_d} \sum_{l=1}^{L_d} \sum_{i=1}^{M_c} \left| \hat{\mathbf{K}}_i^l - \mathbf{K}_i \right|, \tag{10}$$

where $\hat{\mathbf{H}}_i$ denotes the output similarity heatmap and $\mathbf{H}_i$ is the ground-truth heatmap. $\hat{\mathbf{K}}_i^l$ is the output keypoint location from the Graph Transformer layer $l$ and $\mathbf{K}_i$ is the ground-truth location. By our framework 3 and internal modules 3.2, our overall training objective is as follow:

$$\mathcal{L} = \mathcal{L}_{pred} + \gamma \cdot \mathcal{L}_{\text{VAE}}, \tag{11}$$

where $\gamma = 1e^{-3}$ is the hyper-parameter. During inference, our model uses the final layer output $\hat{\mathbf{K}}_i^{L_d}$ as the predicted location. Within the i-SVAE variational inference process, the latent code $z$ is equal to the mean $\mu$ of the approximate posterior, effectively collapsing the stochastic sampling. This deterministic substitution ensures stable and consistent structural priors, aligning with the learned feature distribution while eliminating inference-time uncertainty.

## 4 EXPERIMENTS

### 4.1 IMPLEMENTATION DETAILS

We train and evaluate our method on a machine with an NVIDIA A100 GPU with 40 GB of memory. The architecture is implemented within the MMPose framework Contributors (2020). To ensure a fair comparison, the configuration settings remain consistent with GraphCape Hirschorn & Avidan (2024) and CapeFormer Xu et al. (2022a). During training, we use $256 \times 256$ input images and apply data augmentation including random scaling in the range ($[-0.15, 0.15]$) and random rotation within ($[-15°, 15°]$) for fair comparisons. All models are trained for 200 epochs with a step-wise learning rate scheduler that decreases by a factor of 10 at the 160th and 180th epochs. We use Adam optimizer to train the model for 200 epochs with a batch size of 16. See Section A.2 for details.

Table 1: **Comparisons on MP-100:** PCK@0.2 performance under the 1-shot setting. GenCape achieves the best average performance on the average of all splits, outperforming state-of-the-art methods under all three support sets types.

| Type | Method | Support | Split 1 | Split 2 | Split 3 | Split 4 | Split 5 | Avg. |
|---|---|---|---|---|---|---|---|---|
| Image-support | POMNet Xu et al. (2022a) | Image | 84.23 | 78.25 | 78.17 | 78.68 | 79.17 | 79.70 |
| | CapeFormer Shi et al. (2023) | Image | 89.45 | 84.88 | 83.59 | 83.53 | 85.09 | 85.31 |
| | ESCAPE Nguyen et al. (2024) | Image | 86.89 | 82.55 | 81.25 | 81.72 | 81.32 | 82.74 |
| | MetaPoint+ Chen et al. (2024) | Image | 90.43 | 85.59 | 84.52 | 84.34 | 85.96 | 86.17 |
| | CapeFormer-T Shi et al. (2023) | Image | 89.48 | 86.69 | 85.31 | 84.79 | 84.97 | 86.25 |
| | SDPNet (HRNet-32) Ren et al. (2024) | Image | 91.54 | 86.72 | 85.49 | 85.77 | 87.26 | 87.36 |
| | SCAPE Liang et al. (2024) | Image | 91.67 | 86.87 | 87.29 | 85.01 | 86.92 | 87.55 |
| Text-support | CLAMP Zhang et al. (2023) | Text | 72.37 | - | - | - | - | - |
| | X-Pose Yang et al. (2024) | Image\Text | 89.07 | 85.05 | 85.26 | 85.52 | 85.79 | 86.14 |
| | PPM+CPT Peng et al. (2024) | Image + Text | 91.03 | 88.06 | 84.48 | 86.73 | 87.40 | 87.54 |
| | CapeX-S Rusanovsky et al. (2025) | Image + Text + Graph | 95.17 | 88.88 | 87.72 | 88.24 | 91.81 | 90.37 |
| Graph-support | GraphCape-T Hirschorn & Avidan (2024) | Image + Graph | 91.19 | 87.81 | 85.68 | 85.87 | 85.61 | 87.23 |
| | **GenCape-T (Ours)** | Image (Graph) | **92.05** | **88.69** | **86.89** | **85.88** | **87.02** | **88.09** |
| | GraphCape-S Hirschorn & Avidan (2024) | Image + Graph | 94.73 | 89.79 | **90.69** | 88.09 | 90.11 | 90.68 |
| | **GenCape-S (Ours)** | Image (Graph) | **95.23** | **90.60** | 89.46 | **89.32** | **90.43** | **91.01** |

Table 2: **Performance comparisons** under 5-shot setting with SwinV2-small as image backbone.

| Method | Split 1 | Split 2 | Split 3 | Split 4 | Split 5 | Avg. |
|---|---|---|---|---|---|---|
| Fine-tune | 71.67 | 57.84 | 66.76 | 66.53 | 60.24 | 64.61 |
| POMNet | 84.72 | 79.61 | 78.00 | 80.38 | 80.85 | 80.71 |
| CapeFormer | 91.94 | 88.92 | 89.40 | 88.01 | 88.25 | 89.30 |
| SDPNet | 93.68 | 90.23 | 89.67 | 89.08 | 89.46 | 90.42 |
| PPM+CPT | 93.64 | 92.71 | 91.76 | 92.85 | 91.94 | 92.58 |
| SCAPE | 95.18 | 91.25 | 91.78 | 90.74 | 91.10 | 92.01 |
| GraphCape | 96.67 | 91.48 | **92.62** | 90.95 | 92.41 | 92.83 |
| **GenCape (Ours)** | **97.19** | **92.94** | 92.26 | **91.93** | **93.34** | **93.53** |

Table 3: **Performance comparison** of CAPE methods under stricter thresholds.

| Method | Th0.05 | Th0.1 | Th0.15 | Th0.2 | mPCK |
|---|---|---|---|---|---|
| POMNet | 44.39 | 68.87 | 79.39 | 84.23 | 69.22 |
| CapeFormer | 51.03 | 75.17 | 84.87 | 89.45 | 75.13 |
| ESCAPE | 48.24 | 72.25 | 82.30 | 86.89 | 72.42 |
| MetaPoint | 55.08 | 77.12 | 85.81 | 90.43 | 77.11 |
| GraphCape | 48.55 | 73.43 | 83.71 | 88.19 | 73.47 |
| SCAPE | 54.09 | 77.34 | 87.02 | 91.67 | 77.53 |
| FMMP | 57.30 | 78.48 | 87.28 | 91.82 | 78.72 |
| **GenCape-R50** | **58.91** | **80.63** | **88.97** | **92.74** | **80.31** |

## 4.2 DATASET AND METRIC

We train and evaluate our method on the MP-100 Xu et al. (2022a) dataset, which is currently the only public dataset for CAPE tasks. MP-100 contains 100 sub-categories and 8 supercategories, with a total of 18K images and 20K annotations. The number of annotated keypoints covers a wide range, from 8 to 68. To ensure and identify performance stability on unseen categories, following previous methods Xu et al. (2022a), the dataset is divided into 5 splits. In each split, these categories are split into train, validation, and test sets in a 70/10/20 ratio without any category overlap. We use the Probability of Correct Keypoint (PCK) as the evaluation metric. We follow the standard metric PCK@0.2 as the default metric for performance reporting. And we further evaluate model performance under stricter threshold conditions ([0.05, 0.1, 0.15, 0.2]) for comprehensive comparisons.

## 4.3 BENCHMARK RESULTS

We conduct a comparative analysis of our approach with SwinV2 Liu et al. (2022) as backbone, against the **graph-support method:** GraphCape Hirschorn & Avidan (2024) as our baseline; **image-support methods**: POMNet Xu et al. (2022a), CapeFormer Shi et al. (2023), ESCAPE Nguyen et al. (2024), MetaPoint Chen et al. (2024), SDPNet Ren et al. (2024), FMMP Chen et al. (2025a); **text-support methods:** CLAMP Zhang et al. (2023), XPose Yang et al. (2024), PPM+CPT Peng et al. (2024), CapeX Rusanovsky et al. (2025). Our evaluation is conducted on the MP-100 dataset, considering the 1- and 5-shot settings. We denote our models as GenCape-T and GenCape-S for abbreviation, corresponding to employing SwinV2-tiny and small as the image backbone, respectively.

**1-shot results.** As reported in Table 1, GenCape-T achieves an average PCK of 88.09%, surpassing the strong graph-based GraphCape-T baseline (87.23%) by +0.86%. Remarkably, without relying on class-level text, our method still outperforms multimodal CAPE models such as XPose (86.14%) and PPM+CPT (87.54%), indicating that the learned structure-aware representation serves as an effective surrogate for external semantic cues. Moreover, GenCape-S attains 91.01% PCK, exceed-

ing CapeX-S (90.37%), which leverages both textual and skeleton support. **5-shot results.** Under the 5-shot setting (Table 2), GenCape-S further improves, achieving an average PCK of 93.53% and outperforming all representative CAPE methods, including PPM+CPT (92.58%), SCAPE (91.01%), and GraphCape (92.83%). The model delivers consistent gains across all splits except Split 3, with a maximum of 97.19% on Split 1 and a minimum of 91.93% on Split 4.

**More detailed comparisons.** Table 3 presents results under stricter thresholds (0.2, 0.15, 0.1, 0.05) on Split-1 with ResNet-50 as backbone. Threshold choice strongly affects relative gaps: GenCape-R50 outperforms FMMP by 0.92% at PCK@0.2, and the margin increases to 1.61% at PCK@0.05, showing that coarse thresholds may mask fine-grained discriminative ability. GenCape achieves the highest accuracy at all thresholds and improves mPCK by +1.59% over FMMP.

## 4.4 ABLATION STUDY

In this section, we conduct all of the ablation studies of our proposed method on the MP-100 dataset Split-1 using the SwinV2-Tiny backbone, unless otherwise specified. We now present key ablation experiments. Additional ablations can be found in Appendix B.

**Effects of Different Components.** To rigorously quantify the contribution of each module in our framework, we conduct a comprehensive ablation study under 1-shot setting. We isolate and examine the effects of variational regularization ($\mathcal{L}_{\mathrm{KL}}$), sparsity penalization ($\mathcal{L}_{\mathrm{sparse}}$), and the CGT module. Table 4 reports the results. Starting from the baseline that excludes all components, we observe a base PCK of 91.19%. Introducing the KL divergence term $\mathcal{L}_{KL}$ yields a modest improvement (+0.24%),

Table 4: **Ablation studies** on different components (Split-1).

| $\mathcal{L}_{\mathrm{KL}}$ | $\mathcal{L}_{\mathrm{sparse}}$ | CGT | PCK | $\Delta$ |
|---|---|---|---|---|
| | | | 91.19 | 0 |
| ✓ | | | 91.43 | +0.24 |
| ✓ | ✓ | | 91.75 | +0.56 |
| ✓ | ✓ | ✓ | **92.05** | **+0.86** |

suggesting its stabilizing role by constraining posterior distributions with the Gaussian prior. Combining both constraints, the performance improves substantially to 91.75%, confirming their synergistic effect in enforcing informative and interpretable structural cues. Further incorporating the CGT mechanism yields 0.86% gains, underscoring the importance of compositional graph fusion in consolidating uncertainty-aware structural hypotheses and enhancing query-specific robustness.

**Effects of Hyper-parameter.** We investigate the influence of four key hyperparameters: latent dimensionality $D_z$, sample count $N_s$, and loss weights $\beta$ and $\gamma$ (Eq.5, Eq.11). As shown in Table 5, the model achieves optimal performance at $D_z = 32$, with larger dimensions leading to performance degradation (e.g., 92.05% at 32 vs. lower at 64/128), indicating that excessive latent capacity introduces redundancy and weakens structural compactness. Fixing $D_z = 32$, we vary $N_s$ and observe the best results at $N_s = 3$, while both smaller ($N_s = 2$) and larger ($N_s = 5$) values slightly reduce accu-

Table 5: **Ablation studies** on hyper-parameters, including latent dimension $D_z$, number of posterior samples $N_s$, and weighting factors $\beta$ and $\gamma$ for the training objectives.

| Hyper-parameters of i-SVAE | | | |
|---|---|---|---|
| Parameter $D_z$ ($N_s = 3$)    32 | 64 | 96 | 128 |
| PCK    **92.05** | 91.89 | 91.54 | 91.40 |
| Parameter $N_s$ ($D_z = 32$)    2 | 3 | 4 | 5 |
| PCK    91.60 | **92.05** | 91.47 | 91.63 |
| **Objective weighting factor** | | | |
| Parameter $\beta$ ($\gamma = 1e-3$)    1 | $1e^{-1}$ | $1e^{-2}$ | $1e^{-3}$ |
| PCK    91.43 | **92.05** | 91.65 | 91.74 |
| Parameter $\gamma$ ($\beta = 0.1$)    1 | $1e^{-1}$ | $1e^{-2}$ | $1e^{-3}$ |
| PCK    90.99 | 91.14 | 91.71 | **92.05** |

racy. This suggests a trade-off between uncertainty modeling and variance over-smoothing. For loss weights, we separately tune $\gamma$ (KL loss) and $\beta$ (sparsity loss). The model peaks at $\gamma = 10^{-3}$ when $\beta$ is fixed, balancing reconstruction and regularization. Likewise, $\beta = 0.1$ yields the best PCK, highlighting its role in regulating structure sparsity without over-penalizing connectivity. These results confirm that our framework remains robust across hyperparameter variations, with optimal settings jointly promoting expressiveness and structural regularity.

**Effects of Cross-Category Generalization.** To assess structural generalization beyond category boundaries, we conduct three cross-supercategory pairs experiments: Person↔Felidae, Felidae↔Ursidae, and Person↔AnimalFace. These settings cover diverse appearance and topology gaps, including upright-to-quadruped transfers and full-body to face shifts. As shown in Table 6, GenCape consistently outperforms GraphCape across all pairs, with margins up to +11.8 points (*Fe-*

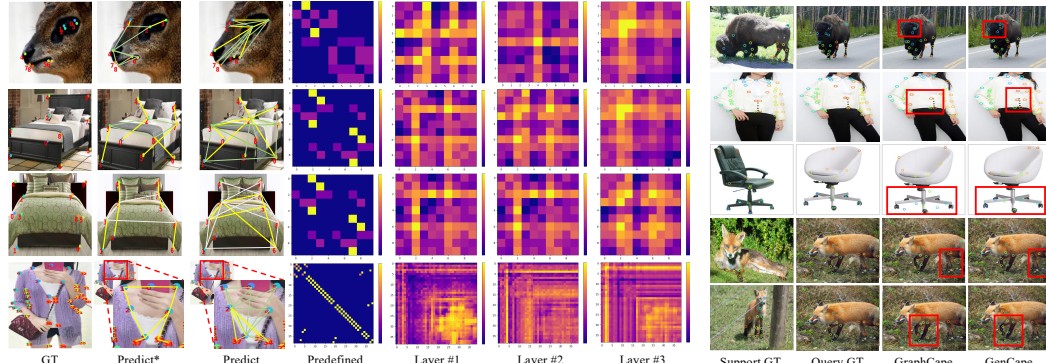

GT    Predict*    Predict    Predefined    Layer #1    Layer #2    Layer #3    Support GT    Query GT    GraphCape    GenCape

Figure 3: Comparisons on adjacency matrices inferred by i-SVAE and predefined graph. The Predict* is the predicted locations with the prior connections, while the Predict is with learned connections.

Figure 4: Comparisons of qualitative visualization. The red boxes highlight the differences.

*lidae→Person*), demonstrating the robustness of compositional latent graphs under structural shifts. We further evaluate in the challenging cross-species setting, where models trained on human bodies and tested on morphologically distinct animal bodies. GraphCape suffers substantial performance drops, e.g., 31.09 on *Rabbit Body*, due to the rigidity of static priors. In contrast, GenCape maintains strong transferability, achieving +21.05 improvement and scaling effectively across species with varying skeletons on *Squirrel Body*. These results validate that generative, instance-specific graphs better capture structural uncertainty and enable robust pose reasoning across categories.

**Qualitative Analysis.** To intuitively understand how our model mines effective skeletons, we visualize the pre-normalized adjacency matrices across decoder layers with the inferred skeletons in Figure 3. Compared with fixed skeletons, whose manually chosen edges are not guaranteed to be semantically correct and can mislead message passing, the graph inferred by i-SVAE are query-conditioned and instance-specific. So pose, viewpoint and other changes are directly reflected in the adjacency matrix. The learned structures are clearly task-driven, emphasizing high-influence keypoints that most contribute to accurate localization, *e.g.*, nose and eyes

Table 6: Cross category evaluation comparing Graph-Cape and GenCape based on SwinV2-small backbone.

| Train | Test | GraphCape | Ours |
|-------|------|-----------|------|
| **Cross super-category** | | | |
| Person | Felidae | 58.46 | **58.82** |
| Felidae | Person | 56.26 | **68.08** |
| Felidae | Ursidae | 73.10 | **74.10** |
| Ursidae | Felidae | 73.83 | **77.76** |
| AnimalFace | Person | 65.49 | **77.93** |
| Person | AnimalFace | 65.08 | **68.41** |
| **Cross species** | | | |
| Human Body | Fox Body | 46.65 | **57.47** |
| Human Body | Rabbit Body | 31.09 | **52.14** |
| Human Body | Squirrel Body | 54.30 | **73.37** |

as anchors for klipspringer face, four corners for bed, and central torso for long sleeved outwear. They receive denser and stronger connections and thus provide more effective geometric constraints. Figure 4 highlights GenCape's superior adaptability when exposed to varying support-query pairs in 1-shot setting. Compared to GraphCape, which fails under pose misalignment and occlusion (*e.g.*, bison and swivelchair), our model consistently localizes keypoints accurately by leveraging its uncertainty-aware graph. Interestingly, we further visualize the impact of varying support images for the same query instance, as shown in the fourth and fifth rows. When the fox undergoes different forms of occlusion, GraphCape suffers from inconsistent errors, while GenCape maintains stable predictions. These results suggest that our method is significantly less susceptible to the adverse effects introduced by suboptimal support sets. Additional analyzes are provided in Appendix C.

## 5  CONCLUSION

In this paper, we introduce GenCape, a generative framework for CAPE that infers keypoint relationships solely from visual inputs. GenCape integrates an iterative Structure-aware Variational Autoencoder to progressively infer instance-specific keypoint relationships, alongside a Compositional Graph Transfer module that aggregates multiple latent graph hypotheses into query-aware structural cues. Extensive experiments on MP-100 demonstrate that GenCape achieves the state-of-the-art performance. These results demonstrate the effectiveness of our framework.

## 6 ACKNOWLEDGMENT

This work was supported in part by New Generation Artificial Intelligence-National Science and Technology Major Project under Grant 2025ZD0123701, in part by the National Key Research and Development Project under Grant 2023YFC3806000, in part by the National Natural Science Foundation of China under Grant 62406226, in part by the Shanghai Municipal Science and Technology Major Project under Grant 2021SHZDZX0100, in part sponsored by Shanghai Sailing Program under Grant 24YF2748700, in part by New-Generation Information Technology under the Shanghai Key Technology R&D Program under Grant 25511103500.

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

APPENDIX

# A  ADDITIONAL IMPLEMENTATION DETAILS

## A.1  DETAILS OF NETWORK STRUCTURE

---

**Algorithm 1** Compositional Graph Transfer (CGT)

---

**Require:** Support keypoint embedding $F_s$, query feature $F_q$
1: **for** $l = 1, \ldots, L_d$ **do**
2:     StructureVAE encoding:
$$[\boldsymbol{\mu}^{(l)}, \log(\boldsymbol{\sigma}^{(l)})] = \text{Enc}(F_s^{(l)})$$
3:     Random sampling ($N_s$ turns)
$$\mathbf{z}^{(l)}|_{n=1}^{N_s} = \boldsymbol{\mu}^{(l)} + \boldsymbol{\sigma}^{(l)} \odot \boldsymbol{\epsilon}$$
4:     StructureVAE decoding:
$$\{\tilde{\mathcal{A}}_n^{(l)}\}_{n=1}^{N_s} = \text{Dec}(\mathbf{z}^{(l)}|_{n=1}^{N_s})$$
5:     Computing confidence:
$$w_n = \frac{1}{\sum_{i=1}^{D_z} \sigma_{n,i}^{(l)} + \epsilon}, \tilde{w}_n = \frac{w_n}{\sum_{m=1}^{N_s} w_m}$$
6:     Bayesian aggregation: $\mathcal{A}_{\text{fused}}^{(l)} = \sum_{n=1}^{N_s} \tilde{w}_n \cdot \tilde{\mathcal{A}}_n^{(l)}$
7:     Update support keypoint embedding:
$$F_s^{(l+1)} = \text{GCN}(F_s^{(l)}, \mathcal{A}_{\text{fused}}^{(l)})$$
8:     Compute Gating Scores: $\alpha^{(l)} = \frac{\text{sim}(\text{Pool}(F_q), \boldsymbol{\mu}^{(l)})}{\sum_{l=1}^{L} \text{sim}(\text{Pool}(F_q), \boldsymbol{\mu}^{(l)})}, \quad L \in [1, L_d]$
9:     Fusion across Layers: $\mathcal{A}_{\text{final}}^{(l)} \leftarrow \sum_{l=1}^{L} \alpha^{(l)} \cdot \mathcal{A}_{\text{fused}}^{(l)}$
10: **end for**

---

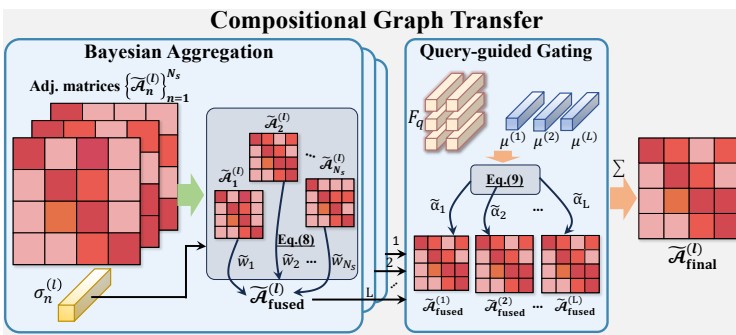

Figure 5: The pipeline of Compositional Graph Transfer. The i-SVAE in the $l$-th decoder layer outputs sampled adjacencies $\{\tilde{\mathcal{A}}_n^{(l)}\}_{n=1}^{N_s}$ with posterior $\mu^{(l)}, \sigma^{(l)}$. Bayesian aggregation fuses samples into a single adjacency estimate $\tilde{\mathcal{A}}_{\text{fused}}^{(l)}$, and query-guided gating reweights layers using the query embedding $F_q$. The result is the final graph $\tilde{\mathcal{A}}_{\text{final}}^{(l)}$ in that layer.

In Table 7, we provide detailed architecture configurations for *i-SVAE* module. $M$ denotes the maximum number of keypoints, $D$ is the node feature embedding size, and hidden_dim = 256 refers to the hidden dimensionality of the VAE encoder. The latent_dim used for graph sampling is denoted by $D_z$, consistent with the main manuscript. Figure 5 illustrates the pipeline of Compositional Graph Transfer. In the $l$-th decoder layer, the i-SVAE produces multiple sampled adjacency matrices $\{\tilde{\mathcal{A}}_n^{(l)}\}_{n=1}^{N_s}$ together with their posterior statistics $\mu^{(l)}$ and $\sigma^{(l)}$. Bayesian aggregation fuses these samples into a single adjacency estimate, and query-guided gating further reweights the fused graphs using the query embedding $F_q$. The resulting matrix $\tilde{\mathcal{A}}_{\text{final}}^{(l)}$ serves as the final structure used for message passing in that layer. The details can be found in Section 3.3.

The complete pipelines of our two proposed modules are outlined in Algorithm 1. Due to the limited structural cues present in raw keypoint features, the iterative learning paradigm faces challenges in capturing meaningful latent graph structures. The resulting graphs $F_s^{(l+1)}$ serve as refined structural

Table 7: Architecture details of the *iterative StructureVAE*, including graph encoder, latent sampling, and graph decoder in the Figure 2 (main manuscript).

| | Norm & Activation | Output Shape |
|---|---|---|
| **Graph Encoder (for posterior distribution)** | | |
| Input node feature $x$ | - | $[B, M, D]$ |
| Reshape $x \rightarrow x'$ | - | $[B, M \cdot D]$ |
| Linear | ReLU | $[B, \text{hidden\_dim}]$ |
| Linear | - | $[B, 2 \cdot \text{latent\_dim}]$ |
| Split $\rightarrow (\mu, \log \sigma^2)$ | - | $[B, \text{latent\_dim}] \times 2$ |
| **Latent Sampling (Reparameterization)** | | |
| If training: $z = \mu + \epsilon \cdot \sigma$ | - | $[B, \text{latent\_dim}]$ |
| Else: $z = \mu$ | - | $[B, \text{latent\_dim}]$ |
| **Graph Decoder (to soft adjacency matrix)** | | |
| Linear | Sigmoid | $[B, M \cdot M]$ |
| Reshape | - | $[B, M, M]$ |
| Symmetrization | $A = \frac{1}{2}(A + A^\top)$ | $[B, M, M]$ |

priors, facilitating the downstream reasoning module to learn more expressive and structure-aware keypoint representations.

## A.2 DETAILED TRAINING CONFIGURATIONS

Table 8 summarizes the training and evaluation settings for the GenCape-T/S models. We adopt the Adam Kingma & Ba (2015) optimizer with a base learning rate of $1.0 \times 10^{-5}$, scheduled linearly and warmed up over 1,000 iterations with a warmup ratio of 0.001. Training is performed for 175,000 epochs with a batch size of 16 in float32 precision, ensuring stable convergence. For few-shot evaluation, the model is assessed under both 1-shot and 5-shot settings, with each episode comprising 15 query samples and a total of 200 episodes to ensure statistical reliability.

## B ADDITIONAL EXPERIMENTAL RESULTS

### B.1 ABLATIONS ON DIFFERENT STRATEGIES

To further assess the effectiveness of our proposed framework, we investigate the impact of different graph construction methods and different compositional graph transfer strategies. Table 9 compares different methods for constructing adjacency matrix. Random initialization results in a significantly performance drop, confirming the importance of structural priors. Learnable graphs consistently outperform static counterparts, and removing the symmetry constraint leads to only a slight decrease, suggesting that bidirectionality is more critical than directional specificity. We replace our layer-wise i-SVAE updates with the first-layer adjacency matrix predicted only once from the encoder output. we observe: iter (92.05) vs. non-iter (91.48). This +0.57 improvement demonstrates that iterative refinement is indeed beneficial. A fixed adjacency estimated once from static support features cannot adapt to the evolving decoder representations. Furthermore, we conducted an experiment that directly used the self-

Table 8: Training configuration used for GenCape-T/S.

| **Training recipe:** | |
|---|---|
| optimizer | Adam |
| **Learning hyper-parameters:** | |
| base learning rate | 1.0E-05 |
| learning rate schedule | linear |
| batch size | 16 |
| training steps | 175,000 |
| lr warmup iters | 1,000 |
| warmup ratio | 0.001 |
| warmup schedule | linear |
| data type | float32 |
| norm epsilon | 1.0E-06 |
| **Few-shot testing hyper-parameters:** | |
| shots | 1 / 5 |
| num_query | 15 |
| num_episodes | 200 |

attention weights from Figure 2 as the adjacency matrix. This variant achieves only 89.33 PCK@0.2 (2.72 drop vs. 92.05), indicating that attention-induced "connectivity" fails to provide meaningful structure. This is expected: self-attention mainly captures appearance-driven correlations, lacks uncertainty modeling, especially when support features are ambiguous. Table 10 investigates composi-

tional graph transfer strategies. The combination of query weighting at the layer level and Bayesian fusion at the sampling level achieves the highest 92.05% PCK. In contrast, using the same strategy at both levels (*e.g.*, query weighting for both) yields suboptimal results. We attribute this improvement to the complementary strengths of the two mechanisms: query weighting enables dynamic alignment of structural importance across layers based on the semantic context of the query, while Bayesian fusion effectively mitigates uncertainty introduced by latent graph sampling. Conversely, a mismatch between Bayesian fusion across layers and query weighting across samples performs worse (91.55%). These findings highlight the importance of hybrid fusion strategies that jointly consider semantic relevance and structural reliability. Layer-wise representations encode semantically distinct structural abstractions that benefit from query-adaptive weighting rather than confidence-based averaging. Applying Bayesian fusion at this level can obscure semantically salient but uncertain layers, effectively flattening meaningful hierarchical distinctions.

Table 9: Comparisons of different adjacency matrix construction strategies under 1-shot setting with SwinV2-Tiny backbone.

| Type | Symmetric | PCK |
|---|---|---|
| Static Graph | ✓ | 91.19 |
| Learnable Graph | ✓ | **92.05** |
| Learnable Graph | ✗ | 91.71 |
| Random Initialized Graph | ✓ | 84.39 |
| Non-iter | ✓ | 91.48 |
| Self-attention | ✓ | 89.33 |

Table 10: Comparisons on compositional graph transfer strategies under 1-shot setting with SwinV2-Tiny backbone.

| Layer-wise | Sampling-wise | PCK |
|---|---|---|
| Bayesian Fusion | Bayesian Fusion | 91.36 |
| Query Weighting | Query Weighting | 91.74 |
| Bayesian Fusion | Query Weighting | 91.55 |
| Query Weighting | Bayesian Fusion | **92.05** |

## B.2 ADDITIONAL CROSS-SUPERCATEGORY RESULTS

Table 11: **Cross-supercategory results.** PCK@0.2 performance under the 1-shot setting on Split-1. Following the standard super-category partitioning protocol, our method achieves the best performance across all splits, demonstrating its strong generalization.

| Method | HumanBody | HumanFace | Vehicle | Furniture |
|---|---|---|---|---|
| ProtoNet | 37.61 | 57.80 | 28.35 | 42.64 |
| MAML | 51.93 | 25.72 | 17.68 | 20.09 |
| Fine-tune | 52.11 | 25.53 | 17.46 | 20.76 |
| POMNet | 73.82 | 79.63 | 34.92 | 47.27 |
| CapeFormer | 83.44 | 80.96 | 45.40 | 52.49 |
| GraphCape | 88.38 | 83.28 | 44.06 | 45.56 |
| **GenCape** | **89.69** | **93.76** | **47.74** | **66.63** |

We follow prior works Liang et al. (2024); Hirschorn & Avidan (2024); Rusanovsky et al. (2025) and perform a cross–supercategory evaluation to rigorously assess the generalization ability of our model across semantically diverse object classes. Concretely, in Table 11 we treat one of the four MP-100 supercategories—**HumanBody, HumanFace, Vehicle, Furniture**—as the test domain and train on the remaining three, creating four disjoint train–test splits. GenCape consistently achieves the highest accuracy across all cross–supercategory splits. These improvements highlight the strong generalization of our generative structural modeling, which captures keypoint dependencies that remain robust under large variations.

We further evaluate a more challenging setting: training on one category and testing on a different, structurally mismatched category. Across all Train→Test pairs, Table 12 shows that GenCape consistently surpasses GraphCape, indicating stronger resilience to cross-category discrepancies. Notably, GenCape achieves substantial gains in HumanBody→HumanFace (+11.36), Vehicle→HumanBody (+8.40), and Chair→HumanBody (+4.23), showing its ability to adapt to entirely different topologies. Overall, these results highlight that GenCape learns transferable, generative keypoint relations that generalize reliably across heterogeneous object categories.

Table 12: **Cross domain transfer evaluation.** PCK@0.2 performance under the 1-shot setting on Split-1. Training on one super-category and testing on the other.

| Train | Test | GraphCape | GenCape |
|-------|------|-----------|---------|
| HumanBody | HumanFace | 33.87 | **45.23** |
| HumanFace | HumanBody | 55.90 | **56.43** |
| Furniture | HumanBody | **73.79** | 73.09 |
| HumanBody | Furniture | 31.11 | **50.47** |
| Vehicle | HumanBody | 50.13 | **58.53** |
| HumanBody | Vehicle | 28.52 | **32.64** |
| Chair | HumanBody | 49.10 | **53.33** |

## B.3 ADDITIONAL METRICS RESULTS

Table 13: **Additional Metrics.** AUC, EPE and NME performance under 1-shot setting.

| Method | AUC % (↑) | EPE (↓) | NME (↓) | PCK % (↑) |
|--------|-----------|---------|---------|-----------|
| GraphCape-T | 89.10 | 41.04 | 0.08 | 91.19 |
| **GenCape-T** | 89.50 | 39.65 | 0.08 | 92.05 |
| GraphCape-S | 91.16 | 30.05 | 0.06 | 94.73 |
| **GenCape-S** | 91.37 | 29.62 | 0.06 | 95.23 |

We further evaluate our model using three standard keypoint localization metrics, as summarized in Table 13. AUC measures the area under the PCK curve and reflects overall accuracy across a range of distance thresholds. EPE computes the Euclidean distance between predicted and ground-truth keypoints. NME reports the mean localization error normalized by object scale. These metrics provide a comprehensive assessment of both absolute and scale-invariant localization performance.

## B.4 COMPARISONS ON COMPUTATIONAL COMPLEXITY

Table 14: Comparison of computational complexity and accuracy across methods.

| Method | GFLOPs | Params | FPS | PCK |
|--------|--------|--------|-----|-----|
| POMNet | 38.01 | 48.21M | 6.80 | 46.05 |
| One-Stage | 22.65 | 26.86M | 36.90 | – |
| CapeFormer | 23.68 | 31.14M | 26.09 | 89.45 |
| GraphCape-T | 15.48 | 43.68M | 15.36 | 91.19 |
| **GenCape-T** | 15.66 | 44.47M | 14.89 | 92.05 |
| GraphCape-S | 27.75 | 65.06M | 10.45 | 94.73 |
| **GenCape-S** | 27.93 | 65.85M | 10.44 | 95.23 |

To further clarify computational efficiency, we provide a comparison of GFLOPs, parameter counts, and inference speed. GraphCape and GenCape are tested on A100, while the results of POMNet, One-Stage, and CapeFormer are taken from CapeFormer Shi et al. (2023) where all measurements were obtained on a RTX 3090. As shown in Table 14, despite this hardware discrepancy, the comparison still reveals a clear trend: GenCape introduces negligible computational overhead relative to GraphCape (e.g., +0.18 GFLOPs and +0.8M params for the Tiny variant), while consistently delivering higher accuracy. This confirms that our generative structural modeling improves performance without sacrificing efficiency.

## C ADDITIONAL VISUALIZATION RESULTS

In this section, we present more qualitative results. As shown in the first row of Figure 6, the support image depicts an inverted top-view of a dog, where GraphCape Hirschorn & Avidan (2024) exhibits

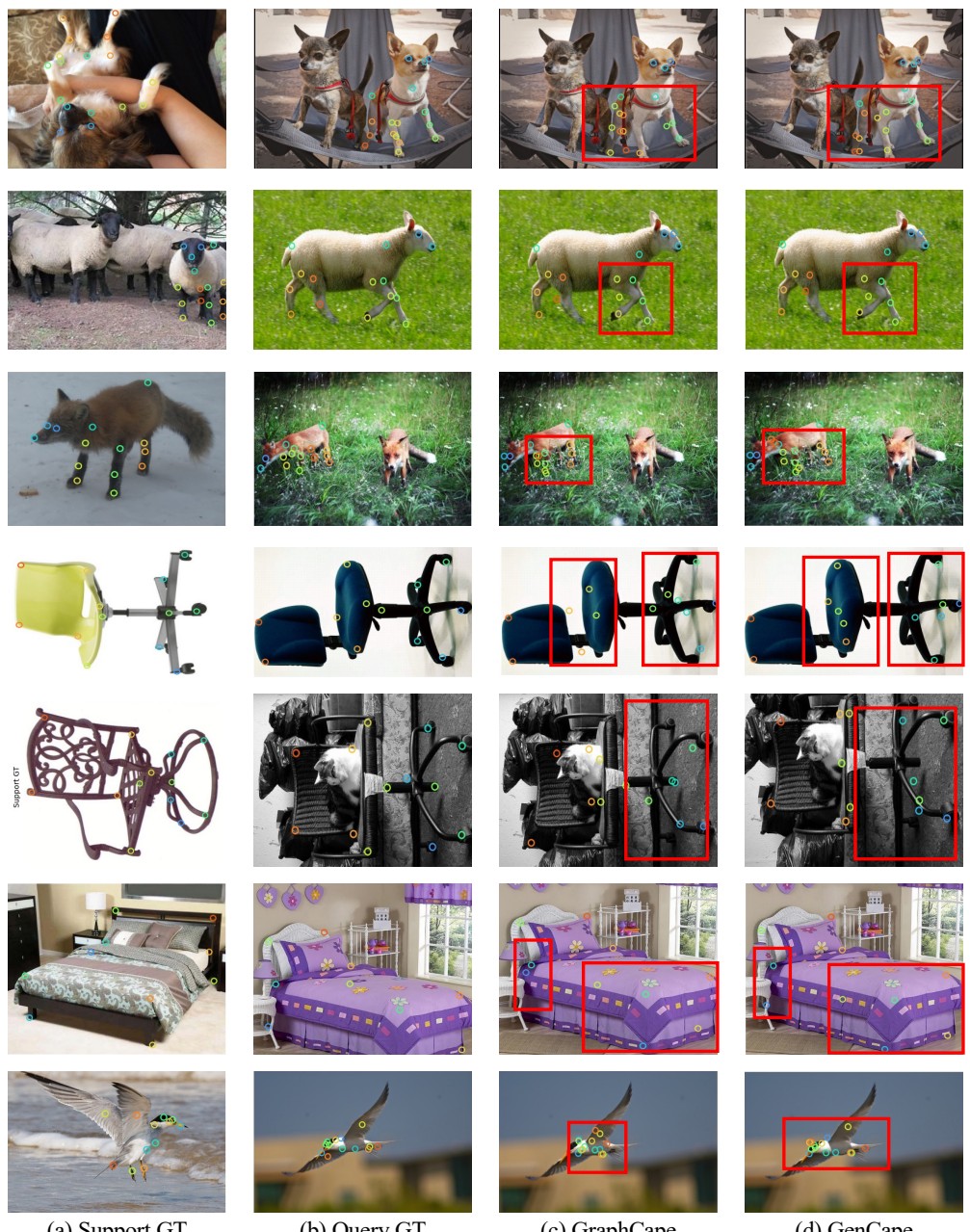

|         |         |         |         |
|---------|---------|---------|---------|
| (a) Support GT | (b) Query GT | (c) GraphCape | (d) GenCape |

Figure 6: **Comparative qualitative results.** We compare more keypoint predictions with GraphCape Hirschorn & Avidan (2024) under the 1-shot setting. The red boxes highlight the regions with significant differences.

noticeable prediction drift on the left hind leg and right foreleg, while our method achieves more accurate localization. In the second and third rows, GraphCape overly relies on support instances with dissimilar poses, resulting in incorrect keypoint predictions. The fourth row presents a challenging case involving a swivel chair, where structural reasoning becomes critical for precise keypoint inference. Despite preserving the overall skeleton shape, GraphCape relies solely on manually defined connections and fails to localize the seat correctly, producing an upward shift as highlighted by the red box. A similar issue is observed in the fifth row, where the predicted seat location is misaligned to the edge of the cat. Figure 7 provides additional category-agnostic pose estimation examples to further illustrate the effectiveness of our approach.

Figure 8 shows more visualizations between predefined skeletons and the latent adjacency matrices inferred across different decoder layers. In the first example (giraffe), the learned adjacency matrices

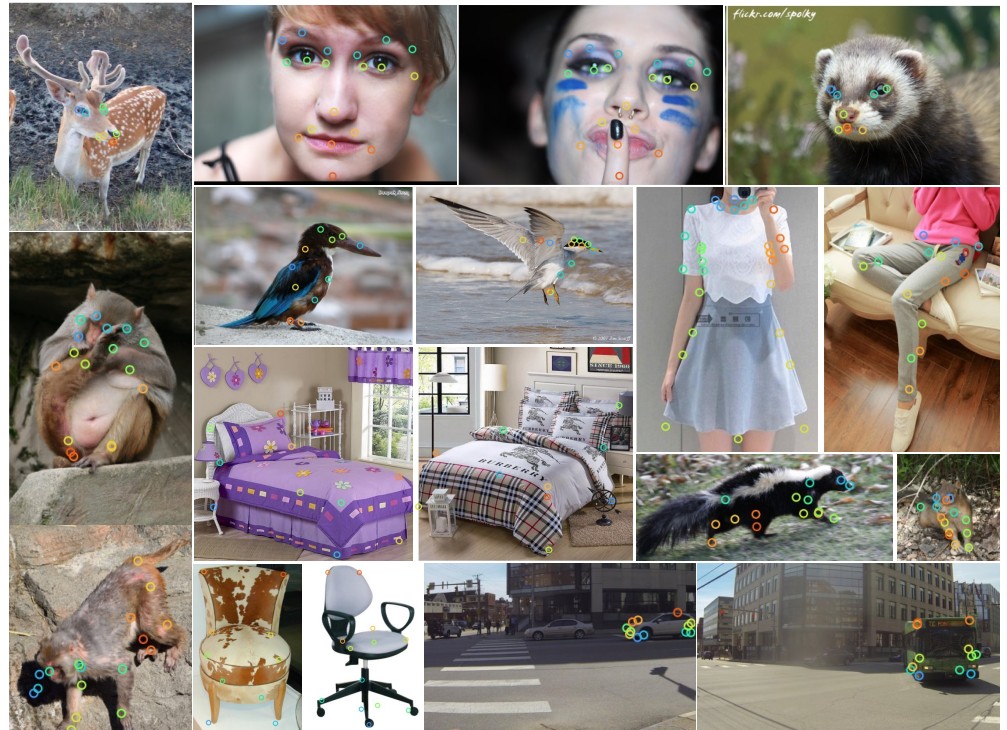

Figure 7: Qualitative visualization. We visualize more keypoint predictions across different splits.

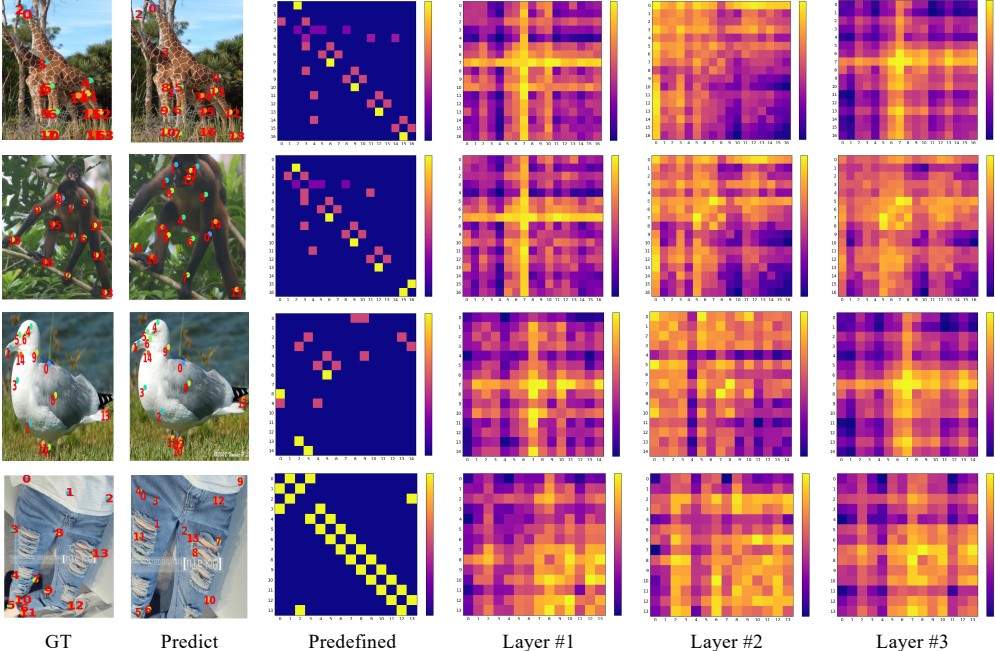

GT              Predict          Predefined          Layer #1          Layer #2          Layer #3

Figure 8: Adjacency matrix visualization. We visualize more latent graph structures across different splits.

progressively capture long-range dependencies critical for reasoning along the elongated neck and legs, outperforming the sparse predefined skeleton. The second case (monkey) presents occlusion challenges from tree branches, where the model adaptively strengthens cross-limb connections in deeper layers to improve structural consistency, though minor prediction drift remains. For the bird, although predefined structures already offer reasonable symmetry, latent graphs further refine bilateral dependencies, enhancing accuracy. The trouser case represents a particularly challenging case due to its dense, repetitive keypoints and weak structural cues. These visualizations also enhance

the interpretability of our GenCape. However, from Figure 4 and 9, we observe that localization errors are primarily caused by visual feature ambiguity. Predictions on the swivelchair category show large structural deviations, suggesting that the weak and homogeneous textures of this class hinder the transfer of support-driven structural priors. We further evaluate robustness by randomly masking the query image. When 25% of the image is masked, GenCape-S shows only a mild drop (93.05), but performance collapses at 50% masking (76.81), showing that heavy occlusion severely disrupts the visual evidence required for localization. This sharp degradation reinforces the need for generative, uncertainty-aware structural modeling to cope with missing keypoints and support–query mismatch.

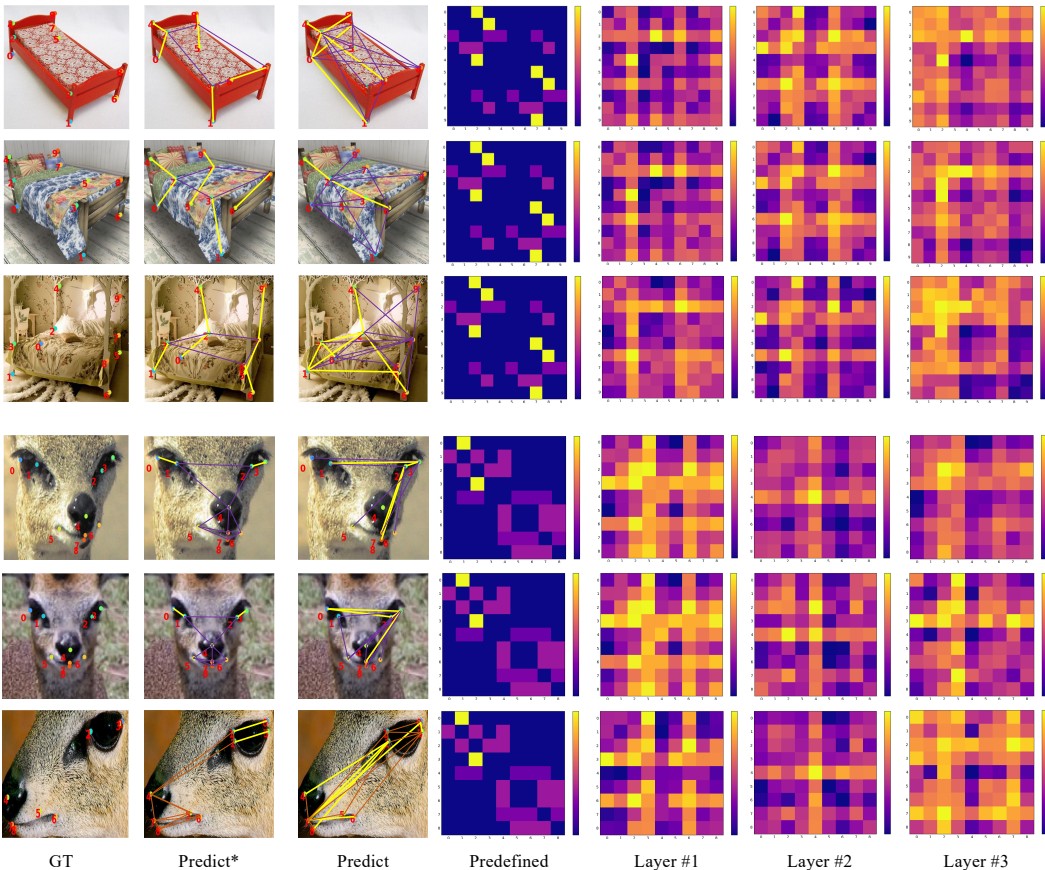

GT          Predict*          Predict          Predefined          Layer #1          Layer #2          Layer #3

Figure 9: **Additional adjacency matrix and skeleton visualization.** We visualized samples from two categories (bed and klipspringer face) in Split-1 for a more comprehensive illustration. The Predict* column is the predicted locations with the prior connections, while the Predict column is with learned connections.

**More qualitative analysis of the adjacency matrix of the same category.** Figure 9 presents additional adjacency-matrix and skeleton visualizations for two same categories in Figure 3. Each row corresponds to the layer-wise adjacency matrices progression produced and finally the skeleton. Brighter colors in the matrices indicate stronger relational dependencies between corresponding keypoints. We observe that adjacency patterns remain similar across samples of the same category. For instance, in first 3 rows of bed category, the adjacency matrices consistently shows a core keypoint showing strong influence on the others. From left to right: initially the core keypoint fires broadly, then adjacent points start to dominate local neighborhoods, and the final layer produces localized high-response clusters and suppressed irrelevant edges, indicating a confident and discriminative dependency graph. For the klipspringer face category in the last three rows, the progression follows a different pattern: early layers emphasize local smoothness among neighboring keypoints, then the model increasingly attends to the central nose region as an anchor, and the final layer converges to a distinct pattern. These differences across categories reflect that the layer-wise graphs represent a coarse-to-fine refinement of functional dependencies. The evidences that the learned graphs encode structural dependencies useful for CAPE, rather than merely aiding optimization convergence.

