# OpenReview forum: "GenCape: Structure-Inductive Generative Modeling for Category-Agnostic Pose Estimation"
_ICLR.cc/2026/Conference — ICLR 2026 Poster_

### Official Review · Reviewer_cNF7 · 2025-10-15

**Soundness:** 4
**Presentation:** 4
**Contribution:** 3
**Rating:** 6
**Confidence:** 4

**Summary:**

This paper introduces GenCape, a novel generative framework for Category-Agnostic Pose Estimation (CAPE) that learns structural relationships directly from support images. The key innovation lies in automatically inferring keypoint connectivity patterns (soft adjacency matrices) without requiring predefined skeleton graphs, keypoint identifiers, or text descriptions. The framework comprises two main components: The Iterative Structure-aware Variational Autoencoder (i-SVAE) learns instance-specific graph structures from support features using variational inference, with iterative refinement across decoder layers. The Compositional Graph Transfer (CGT) module then dynamically combines multiple graph hypotheses through Bayesian fusion and query-guided attention mechanisms.

**Strengths:**

This is the first CAPE method to achieve fully automatic learning of structural relationships from image support sets, removing the need for predefined skeletons, keypoint IDs, or text descriptions, which enhances both generality and practical deployment. The i-SVAE approach models structural uncertainty through variational inference, demonstrating superior robustness compared to discriminative methods like SDPNet, particularly when handling support-query mismatches or occlusion scenarios. The method consistently outperforms various baselines on MP-100, with particularly notable advantages under strict thresholds (e.g., PCK@0.05).

**Weaknesses:**

1. While the paper claims to evaluate cross-supercategory generalization on MP-100, the definition of supercategories appears inconsistent with the original MP-100 benchmark and prior CAPE literature (e.g., CapeFormer).
The original MP-100 dataset is widely understood to group categories into four high-level semantic domains: human body, human/animal face, vehicle, and furniture. However, this work instead uses a finer-grained 8-supercategory split (e.g., separating Felidae, Canidae, Ursidae as distinct supercategories), which blurs the line between "cross-category" and "cross-subcategory" generalization. For instance, transferring from Felidae to Ursidae involves structurally similar quadruped animals with comparable keypoint layouts—this is arguably intra-domain transfer, not the more challenging cross-domain shift (e.g., chair → human) that truly tests category-agnostic capability. Worse, the paper does not include any cross-domain transfer between the canonical four domains (e.g., furniture → human body). This omission is critical, as if the method cannot generalize from chair to person, its claim of "structure-inductive" modeling is significantly weakened.

2. The paper does not provide any comparison of computational efficiency, such as inference time, FLOPs, model size, or throughput, against baseline methods like GraphCape or CapeFormer. While it introduces additional modules (i-SVAE and CGT) that likely increase computational cost, no quantitative analysis or efficiency trade-offs are reported.

**Questions:**

Figures 4 and 5 reveal some localization errors. What is the primary cause of these errors, structural inference failures or visual feature ambiguity?

Could the authors provide quantitative analysis of these failure modes?

What is the method's robustness to scale variations, cropping, and other common transformations?

Have additional evaluation metrics beyond PCK been considered, such as AUC or other standard pose estimation metrics?

---

> ### Author Response · Authors · 2025-11-21
>
> We sincerely thank the reviewer for the positive recognition of our work and for the constructive and insightful suggestions, which help make our contribution more solid. Below, we address each question and concern in detail.
>
> **Weakness 1**. Thank you for raising this important point. Our supercategory split indeed follows the **'supercategory' field defined in the original MP-100 annotation file**, which is not fully align with the four-supercategory (HumanBody, HumanFace, Vehicle, and Furniture) setting commonly used in CapeFormer and prior works. To address this concern and ensure a comparable evaluation we
> have additionally conducted the experiments.
> We designate one of the four supercategories in the MP-100 dataset—human face, human body, vehicle, or furniture—as the test set, while the remaining categories are used for training.
> We follow the same setting as the main paper in Section 4.3, and re-implement GraphCape.
> The results are reported in Table R2.
>
> **Table R2**: Cross super-category pose estimation results. PCK@0.2 performance under the 1-shot setting on split 1.
> | **Method**   | **HumanBody** | **HumanFace** | **Vehicle** | **Furniture** |
> |--------------|----------------|----------------|--------------|----------------|
> | ProtoNet     | 37.61          | 57.80          | 28.35        | 42.64          |
> | MAML         | 51.93          | 25.72          | 17.68        | 20.09          |
> | Fine-tune    | 52.11          | 25.53          | 17.46        | 20.76          |
> | POMNet       | 73.82          | 79.63          | 34.92        | 47.27          |
> | CapeFormer   | 83.44          | 80.96          | 45.40        | 52.49          |
> | GraphCape    | 88.38          | 83.28          | 44.06        | 45.56          |
> | **GenCape**      | **89.69**          | **93.76**          | **47.74**        | **66.63**          |
>
> Furthermore, as suggested by the reviewer, we conducted cross-domain experiments on the four canonical domains as well as the challenging **chair → HumanBody** setting. We followed the experimental setup of Section 4.3 and only reduced the training epochs to 50. The results are shown below:
>
> Table R3: Cross domain transfer results. PCK@0.2 performance under the 1-shot setting.
> | **Train**     | **Test**      | **GraphCape** | **Ours** |
> |---------------|---------------|---------------|----------|
> | HumanBody     | HumanFace     | 33.87         | **45.23**    |
> | HumanFace     | HumanBody     | 55.90         | **56.43**    |
> | Furniture     | HumanBody     | **73.79**         | 73.09    |
> | HumanBody     | Furniture     | 31.11         | **50.47**    |
> | Vehicle       | HumanBody     | 50.13         | **58.53**    |
> | HumanBody     | Vehicle       | 28.52         | **32.64**    |
> | chair         | HumanBody     | 49.10         | **53.33**    |
>
> The results in Table R3 show a clear and consistent improvement over GraphCape across nearly all transfer types. The gains are most pronounced in the challenging cross-domain shifts (e.g., HumanBody → Furniture, chair → HumanBody), indicating that GenCape provides a more
> robust and transferable structural prior. Even when GraphCape performs strongly (e.g., Furniture → HumanBody), our method matches its performance. Overall, the improvements validate that GenCape enhances structural robustness and effectively handles cross-domain shifts that previous CAPE approaches fail to generalize to.
>
> **Weakness 2**. We appreciate the suggestions. To further clarify computational efficiency, we provide a comparison of GFLOPs, parameter counts, and inference speed:
>
> **Table R4**: Comparison of efficiency and accuracy. We test the efficiency under 1-shot setting.
> | **Method**              | **GFLOPs** | **Params** | **FPS** | **PCK** |
> |------------------------|-----------:|-----------:|--------:| --------:|
> | POMNet                 | 38.01      | 48.21 M     | 6.80   | 46.05 |
> | One-Stage              | 22.65      | 26.86 M    | 36.90   | - |
> | CapeFormer             | 23.68      | 31.14 M    | 26.09   | 89.45 |
> | GraphCape-Tiny         | 15.48      | 43.68 M    | 15.36   | 91.19 |
> | GraphCape-Small        | 27.75      | 65.06 M    | 10.45   | 94.73 |
> | **GenCape-Swin-Tiny**  | 15.66      | 44.57 M    | 14.89   | 92.05 |
> | **GenCape-Swin-Small** | 27.93      | 65.85 M    | 10.44   | 95.23 |
>
> These results show that GenCape achieves a strong **accuracy–efficiency trade-off**:
> it delivers notable PCK improvements over GraphCape while keeping GFLOPs, parameter
> counts, and FPS essentially unchanged (e.g., 15.48 → 15.66 GFLOPs and 15.36 → 14.89 FPS for the Tiny models).

---

> ### Author Response · Authors · 2025-11-21
>
> **Question 1**. From Figures 4 and 5, we find that the **primary cause of the localization errors is visual feature ambiguity**, while structural inference failures play a secondary role. First, predictions on the **swivelchair** category exhibit large structural deviations, indicating that the weak texture on this class make it difficult for support-driven structural priors to transfer. Second, when **occlusion** is present, the localization error becomes much more pronounced, highlight the importance of generative and uncertainty-aware structural modeling to overcome missing keypoints.
>
> **Question 2**. We provide quantitative results for both modes, as summarized in the Table R5.
>
> **Table R5**: Performance comparison on the challenging *swivelchair* category.
> | **Test swivelchair** | **AUC %(↑)** | **EPE (↓)** | **NME (↓)** | **PCK %(↑)** |
> |------------------|-----------|---------|---------|-----------|
> | GraphCape-T | 79.39     | 110.48   | 0.18    | 63.18     |
> | GenCape-T | 85.93     | 72.29   | 0.11    | 83.81     |
>
> We evaluate the trained GenCape-T and GraphCape-T model on the swivelchair category.
> Our GenCape outperform significantly than GraphCapeon this challenging category.
> This phenomenon can be explained by the characteristics of the swivelchair category: it contains weak textures, smooth surfaces, and highly variable geometric layouts. Such instances rely far more on **structural understanding** than on appearance cues. As a result, discriminative feature–based methods like GraphCape tend to fail when texture signals are insufficient.
> And we also observe that is indeed noticeably lower than overall performance (83.81 vs. 92.05). This category exhibits uncommon part configurations, and weak appearance cues, making it a particularly challenging case for category-agnostic pose estimation.
>
> **Table R6**: Masking the query image. Performance under different masking percentage.
> | **Masking Setting** | **AUC %(↑)** | **EPE (↓)** | **NME (↓)** | **PCK %(↑)** |
> |----------------------|-----------|---------|---------|-----------|
> | random mask 25%      | 90.50     | 33.95   | 0.07    | 93.05     |
> | random mask 50%      | 83.94     | 67.67   | 0.13    | 76.81     |
>
> For the second failure mode, **occlusion**, we also conducted GenCape-S experiments by randomly masking regions of the query image to simulate missing visual evidence.
> We observe that when only a small portion of the query image is masked, the performance drops only mildly. However, as the masking ratio increases, the model’s accuracy collapses rapidly, indicating that heavy occlusion severely disrupts the visual cues required for reliable keypoint localization.
>
> **Question 3**. To evaluate the robustness of our method under common transformations, we conducted controlled tests with different combinations of **scaling** and **rotation** applied to the query images. As shown in the Table R7:
>
> **Table R7**: Performance under different query image transformations.
> | **Transformations**   | **PCK** |
> |--------------------|----------------------|
> | 0 scale, 0 rot    | 92.05                 |
> | 0 scale, 15 rot    | 91.20                |
> | 15 scale, 0 rot   | 91.32                 |
> | 30 scale, 0 rot   | 91.56                 |
> | 0 scale, 30 rot   | 90.51                 |
>
> The results show that GenCape remains highly robust under moderate scaling and rotation, with only slight performance degradation across all tested settings. We also observe that GenCape is **more sensitive to rotation than to scale changes**: for the same magnitude of perturbation, rotational transformations introduce a larger drop.
>
> **Question 4**.
> We evaluate our model using additional three standard keypoint localization metrics, and the results are reported in Table R8.
> **(1) AUC (↑)** Area Under Curve, is defined as the area under the PCK curve and reflects overall accuracy across
> multiple distance thresholds;
> **(2) EPE (↓)** Endpoint Error, is defined as the Euclidean distance between the ground-truth and predicted points;
> **(3) NME (↓)** Normalized Mean Error, is defined as the mean localization error of the normalized object scale.
>
> **Table R8**:  Additional Metrics. AUC, EPE and NME performance under 1-shot setting.
> | **Method**  | **AUC %(↑)** | **EPE (↓)** | **NME (↓)** | **PCK %(↑)** |
> |-------------|-----------|---------|---------|-----------|
> | GenCape-T   | 89.50     | 39.65   | 0.08    | 92.05     |
> | GenCape-S   | 91.37     | 29.62   | 0.06    | 95.23     |

---

> ### Comment · Reviewer_cNF7 · 2025-11-27
>
> Thank you for the rebuttal and the additional experiments. Most of my concerns have been adequately addressed, particularly regarding the evaluation protocol, computational efficiency, and robustness to common transformations. The new results in Tables R2–R8 provide strong support for the method’s effectiveness and generalization capability.

---

### Official Review · Reviewer_5bdU · 2025-10-22

**Soundness:** 4
**Presentation:** 3
**Contribution:** 3
**Rating:** 6
**Confidence:** 4

**Summary:**

This works suggests solving CAPE by progressivley inferring instance-specific keypoint relationships from the support, instead of using predefined annotated adjacency matrices. The authors also introduce the Compositional Graph Transfer module that aids with incoroporating the query features, thus allowing for less reliance on inferred keypoint relationships from the support. This makes the model more robust to occlusions and discepencies between the support and query. The new GenCape approach is tested on the known MP-100 benchamrk, achieving SOTA results.

**Strengths:**

1. The paper is written in a clear language that was easy to follow.

2. While predicting the keypoints relations from the data is not new, the novel i-SVAE and CGT components suggest some interesting insights that might interest the CAPE community.

3. The suggested approach achieves SOTA while dropping the need for predefined annotated data (keypoint connectivity) that was used by previous methods.

4. Other than the main experiment in Table 1, the design choices are justified in the ablations conudcted (Table 4, Table 5 and Table 6).

**Weaknesses:**

My main issues are with the presentation, not with the method. After resolving these issues, I would positively consider increasing my rating.

1. The technical text (mostly) in the Methods section:

Line 157: M_C is not defined in the right place. Move it to this sentence.

Line 179: remove 1 between.

Lines 190-195: i-SVAE also infers graphs from the support. And as you mention, there is sometimes a discrepency between the support and the query. So I'm not sure that i-SVAE alone will solve the issue mentioned in these lines. However, i-SVAE combined with CFG will.

Line 208: F_s^(l-1) is not defined properly - what is its value where l=1?

Line 212: in the second row of Equation 1, should this be F_s^(l-1) or F_s^(l)?

Equations 3 and 4: A^~(l) is defined twice?

Equation 6: F_s^(l) is in the input and output

Line 244: A^~(l) - different notation compared to Equation 3. Should be/not be in bold?

Line 248: This is not clear. Do keypoint locations are predicted in each layer? Each layer of what? the Graph Transformer Decoder? Clarify what is the output of each layer in the Graph Transformer Decoder.

Line 275: missing ')' in mu^(l

Line 377: "More detailed comparisons.": should be a sperate paragraph?

2. Figures:

Figure 2: Consider adding CGT to Figure 2 (A^l_fused is not enough to easily follow).

Figure 3: It is challenging to interpret the adjacency matrices. Consider showing the “best” links from the adjacency matrix as colored edges in your prediction.

Figure 4: last column is AutoCape.

**Questions:**

You mentioned Text Graph Support as an approach for CAPE. Will fusing text with your approach might increase performance? Could you hint on how whould you incorporate text as a future work with your approach (maybe also infer it from the support?).

---

> ### Author Response · Authors · 2025-11-21
>
> We sincerely thank the reviewer for their detailed feedback and positive assessment of our novelty and framework.
> We have revised our paper according to your comments.
> Below, we provide point-by-point responses to all weaknesses and questions raised in the review.
>
> **Weakness 1: technical text**.
>
> Line 157: $M_c$ denotes the maximum possible number of keypoints. We haved added it in revised manuscript.
>
> Line 179: We have deleted 1 between.
>
> Lines 190-195: The i-SVAE's role is to provide a *generative, uncertainty-aware* structural prior that is more robust than a deterministic. This **mitigates, but does not eliminate the mismatch** between support and query. The discrepancy is explicitly handled by CGT, whose query-guided fusion adapts the latent graphs to the actual visual evidence of the query. In other words, i-SVAE produces a flexible generative prior, and CGT aligns this prior with the query, and **together they effectively address** the discrepency between the support and the query.
>
> Line 208 & 212: We use $F_s^{(l)}$ to represent support node embedding in the $l$-th layer instead of any $F_s^{(l-1)}$, and $F_s^{(1)}$ is directly obtained by Transformer Encoder output $F_s$. Thus, in the second row of Equation 1, there is $F_s^{(l)}$.
>
> Equation 3 & 4: For clearer presentation, we have explicitly revised the left side of Equation (4) to  $\tilde{\mathcal{A}}_{\text{final}}^{(l)}$.
>
> Equation 6: We have revised the left side of Equation 6 to $F_{s}^{(l+1)}$.
>
> Line 244: We have revised it to ${\tilde{\mathcal{A}}_{\text{final}}^{(l)}}$.
>
> Line 248: We clarify that each  Graph Transformer Decoder layer outputs keypoint positions, following the **intermediate
> supervision** commonly used in CapeFormer, GraphCape. The updated keypoint positions from the last decoder layer are used as the final prediction.
>
> Line 275: Have fixed in revised paper.
>
> Line 377: Thank you for the suggestion. We have placed the "More detailed comparisons..." section as a seperate paragraph to improve clarity and readibility.
>
> **Weakness 2: Figures**.
>
> Figure 2: Revising Figure 2 would require substantial re-layout of the full framework, which is non-trivial at this stage. We have added a visual illustration of **CGT** in the Figure 5 in Appendix to further clarify its design and intuition. If needed, we will update the full framework figure.
>
> Figure 3: For better understanding, we have add skeleton visualization on top of the images in Figure 3 in the latest revised manuscript.
>
> Figure 4: Thank you for the reminding, *AutoCape* was a former name. We have fixed in revised paper.
>
> **Question 1**. Thank you for the interesting suggestion. Incorporating text into our framework is indeed a promising future direction, e.g., CapeX. We view this as a valuable extension and plan to explore multi-modal (image + text) structure inference in future work.

---

> > ### Comment · Reviewer_5bdU · 2025-11-26
> >
> > Thank you for your detailed response.
> >
> > **Lines 190-195**: My original comment contained a typo (CFG instead of CGT). Following the authors' clarification, my concern remains: the authors should consider explicitly stating that both CGT and i-SVAE together address the discrepancy between support and query sets mentioned at the beginning of this paragraph, rather than attributing this solely to i-SVAE as currently written.
> >
> > **Equations 6 & 7**: Following your revision, F_s^(l+1) appears to be defined twice. Please clarify this point.
> >
> > **Figure 2**: Thank you for adding Figure 5. To enhance clarity, please consider providing a more detailed caption that describes the components and pipeline flow illustrated in the figure. Specifically, it would be helpful to remind what the variables represent, and from which components CGT receives these inputs (e.g. F_q, μ_{1,2,...,L}).
> >
> > **Figure 3**: The current presentation does not clarify how to interpret the adjacency matrices. Please: (1) explain what characteristics distinguish a good adjacency matrix from a poor one, and (2) provide criteria by which readers can evaluate whether the presented matrices exhibit expected behavior.
> >
> > My original suggestion was to visualize the strongest connections in the matrix by identifying high-weight cells (e.g., cell [i,j]) and overlaying corresponding edges between nodes i and j on the original image. This visualization would better illustrate the semantic relationships captured by these matrices.

---

> > > ### Author Response · Authors · 2025-11-30
> > >
> > > **Line 190-195:** Now, we have **added explicit statement** at the beginning of this paragraph: In CAPE, the support and query mismatch in visibility, poses and topologies, making structural alignment essential. Our framework addresses this discrepancy jointly with *i-SVAE* that learns layer-wise, instance-specific graphs from support features, and *CGT* that adapts them to the query.
> > >
> > > **Equation 6 & 7:** The two equations **express the same operation**. Equation 6 expands the GCN update step into its explicit formulation, while Equation 7 rewrites this update in a compact functional form. Following Line 244, Equation 7 is included again to emphasize the role of the learned adjacency matrix in structural guidance for message passing.
> > >
> > > **Figure 2:** Thank you for the suggestion. We have **added a clearer illustration of Figure 5** in the Appendix: The pipeline of Compositional Graph Transfer. The i-SVAE in the $l$-th decoder layer outputs sampled adjacencies $\lbrace \tilde{\mathcal A}\_n\^{(l)} \rbrace\_{n=1}\^{N_s}$
> > > together with their posterior statistics $\mu^{(l)}, \sigma^{(l)}$. Bayesian aggregation fuses these samples, and query-guided gating reweights layers using the query embedding $F_q$. The resulting matrix $\tilde{\mathcal{A}}_{\text{final}}^{(l)}$ serves as the final structure used for message passing in that layer.
> > >
> > > **Figure 3:** Thank you for your suggestion, and we have revised the paper. Following your advice, we have redrawn the skeleton visualizations in Figure 3 and Figure 9, where thicker connection indicate larger weights, and any weight below 0.5 is treated as no connection.
> > > Meanwhile, we have also added corresponding analysis.
> > > In Figure 3, compared with fixed skeletons, whose manually chosen edges are not guaranteed to be semantically correct and can mislead message passing, thereby harming localization accuracy, the graph inferred by i-SVAE are query-conditioned and instance-specific.
> > > So pose, viewpoint and other changes are directly reflected in the adjacency matrix.
> > > This allows the model to adapt to variations in pose, viewpoint, and occlusion, rather than forcing all instances to follow the same rigid topology.
> > > Across categories, the learned graphs consistently exhibit **task-driven** behavior, emphasizing the dependencies that are most helpful for accurate keypoint localization under the CAPE setting.
> > > For example, in the first row in the Figure 3, klipspringer face, the model identifies the nose and eyes as **structural anchors**, producing strong long-range edges that remain absent in the predefined facial skeleton.
> > > The base anchors of the two bed examples in the second and third rows differ: the first places more emphasis on the central region, whereas the second highlights the four corner keypoints.
> > > This behaviour is expected, as the learned adjacency captures the dependencies specific to each support–query pair, rather than representing a category-level skeleton.
> > > In long sleeved outwear, the model focuses on the central torso region and lower garment boundary, capturing stable reference points.
> > > They receive denser and stronger connections and thus provide more effective geometric constraints.

---

### Official Review · Reviewer_E4Zn · 2025-10-26

**Soundness:** 3
**Presentation:** 3
**Contribution:** 3
**Rating:** 4
**Confidence:** 5

**Summary:**

The paper suggests a novel CAPE method, which utilizes predicted graph structure for enhanced keypoint localization accuracy.
The method uses a graph VAE formulation to predict the graph, and further implements it iteratively within each decoder layer.
Using CGT, several sampled graphs are combined into a query-aware graph structure that aids in localization.
The authors show competitive results on the MP100 dataset.

**Strengths:**

- The paper suggests a novel method that deals with a limitation of recent graph-based methods.
- The paper is well written, and the proposed solution looks solid and practical.
- SOTA results compared to other CAPE methods on the MP100 dataset.

**Weaknesses:**

- Using only Fs to predict the adjacency matrix suggests that the structure information is embedded in Fs in the first place.
As self-attention can be seen as an all-to-all information sharing mechanism, an explanation of why self-attention can't learn the relevant connections between keypoints should be added or even proven.
Specifically, the authors should explain how the current i-SVAE design adds to the self-attention already in the decoder.

- Iterative Graph Prediction - The suggested method works iteratively, predicting a different adjacency matrix for each decoder layer.
An ablation study, showing why using a different adjacency matrix in each decoder layer, compared to using only one predicted adjacency matrix (using the output features of the encoder, for example), should be presented to support the iterative superiority claim.

- Qualitative skeleton visualization - Figure 3 is hard to understand.
It would be helpful to add the skeleton visualizations on top of the images, and not only show the adjacency matrix.
Maybe the width or opacity can correspond to the weight. It is crucial to make it easier to understand what structure is actually learned.

Small Note:
- Figures 4 and 5 label your method as AutoCape instead of GenCape.

**Questions:**

- CGT - the adjacency matrices are sampled using the predicted mean and variance. Thus, it's not clear to me why each sample has its own mean and variance values (line 263), given that they are sampled from the same distribution.
This is further shown in equation 9, where alpha_n is not dependent on n at all.

- See weaknesses for other questions.

 I'm willing to raise my score if the authors address my concerns.

---

> ### Author Response · Authors · 2025-11-21
>
> We sincerely thank the reviewer for the thoughtful and constructive feedback. We have revised our paper according to your comments. We respond to each concern below and clarify the points that may have caused confusion.
>
> **Weakness 1**. We agree that structural cues are partially encoded in the support features $F_s$, but self attention alone is insuffcient to recover a reliable structural topology. To verify this, we performed an additional experiment where the attention weights in self attention in Figure 2 were used as the adjacency. On split-1 under 1-shot setting, this variant obtains **89.33 PCK@0.2**(↓2.72 vs. GenCape-T), showing that attention-induced “connectivity” does not provide useful structure. Self-attention learns **appearance-driven weights**. These weights do not model uncertainty, nor do it enforce sparsity, all of which are crucial in CAPE where support features may be ambiguous.
> In contrast, i-SVAE treats the adjacency as a **latent generative variable**, learning a distribution over possible structures.
> In summary, self-attention is a discriminative mechanism, while i-SVAE provides a **generative structural prior**, and this fundamental difference is crucial under the CAPE setting.
>
> **Weakness 2**. We conducted the requested ablation by replacing our layer-wise i-SVAE updates with the first-layer adjacency matrix predicted only once from the encoder output. On Split-1 Swin-Tiny, we observe: **iter (92.05) vs. non-iter (91.48)**. This **+0.57 improvement** demonstrates that iterative refinement is indeed beneficial. A fixed adjacency estimated once from static support features cannot adapt to the evolving decoder representations. In contrast, our iterative design updates the latent graph at each layer, allowing the structure to progressively align with higher-level, query-conditioned features.
>
> **Weakness 3**. For better understanding, we have add skeleton visualization on top of the images in Figure 3 in the latest revised manuscript.
>
> **Weakness 4**. We thank the reviewer for catching this oversight. *AutoCape* was the former name of our method, and the labels in Figures 4 and 5 were not updated accordingly. We have corrected all occurrences to **GenCape** in the revised manuscript.
>
> **Question 1**. Our implementation follows the standard VAE formulation. Thus, the mean and variance do not differ across sampling turn. In the revision, we have replaced these symbols with $(\mu^{(l)}, \sigma^{(l)})$ in Line 263. Similarly, in Equation (9), the subscript $n$ on $\alpha_n^{(l)}$ is unnecessary.
> We appreciate the reviewer for pointing out these issues, which helps us refine the mathematical presentation and align the notation with the actual implementation.

---

> > ### Comment · Reviewer_E4Zn · 2025-11-23
> >
> > I thank the authors for their detailed comments and added experiments.
> > While most of my initial concerns were addressed, I still have a major concern remaining:
> >
> > Treating the adjacency matrix as a structural prior implies that the structure should remain fixed (or at least consistent) for the same object. This concept is somewhat contradicted by the use of layer-wise structures.
> > For example, is the progression meant to be coarse-to-fine? I find the current interpretation of the learned adjacency as "structural" difficult to grasp.
> >
> > This is further illustrated in the updated Figure 3. The skeletons in the figure seem to contradict the expected behavior of a structural prior.
> > For instance, in the bed examples, looking at the skeleton alone does not suggest that the graph represents a bed. In fact, the predicted structures of the two bed examples are completely different, while the 3D orientation of them is similar.
> > This implies that the learned prior may not be truly structural, but rather a mechanism that aids in model convergence.
> >
> > As this is the main contribution of the paper, I think it would benefit from more visualizations and explanations.

---

> > > ### Author Response · Authors · 2025-11-24
> > >
> > > We sincerely thank the reviewer for the timely follow-up assessment and the thoughtful clarifications. Below we respond to each part in detail, supported by the additional visualizations in Appendix Figure 9 in revised paper.
> > >
> > > > **Should a structural prior remain fixed for the same object?**
> > >
> > > We clarify that in GenCape, the adjacency atrix **is not** a fixed graph for the same object, but an instance-specific dependency graph that represents which keypoints are most informative for each other under the **current support–query pair**. Thus, the structure in our method is a learned, probabilistic graph conditioned on support features $F_s$. For example, in Figure 3 (first row) and Figure 9 (fourth row), the adjacency matrices of klipspringer_face appear very similar, whereas they differ from those in the last row of Figure 9.
> > >
> > > > **Why do similar beds show different adjacency matrices?**
> > >
> > > Even with similar 3D orientation, two bed instances differ in texture, support–query mismatch and et al. Because i-SVAE learns $q(z|F_s)=\mathcal{N}(\mu,\mathrm{diag}(\sigma^2))$, changes in features lead to different uncertainty, and hence different adjacency maps. Similarly, as discussed in our response to Reviewer Dn8o's weakness 3, our analysis of the latent embeddings shows that they form a highly cohesive cluster (ensuring consistency), while still preserving non-trivial variability (capturing diversity).
> > > In other words, even within the same object category, the most informative keypoint relations are **instance-dependent**.
> > >
> > > > **Does layer-wise progression correspond to coarse-to-fine structure?**
> > >
> > > Yes, this is exactly what the layer-wise design achieves, and the updated visualizations in Figure 9 confirm it. For instance, in first 3 rows of bed category in Figure 9, the adjacency matrices consistently shows a **core keypoint** showing strong influence on the others. From left to right: initially the core keypoint fires broadly, then adjacent points start to dominate local neighborhoods, and the final layer produces localized high-response clusters and suppressed irrelevant edges, indicating a confident and discriminative dependency graph. For the klipspringer_face category (last three rows in Figure 9), the progression follows a different pattern: early layers emphasize local smoothness among neighboring keypoints, then the model increasingly attends to the central nose region as an anchor, and the final layer converges to a distinct pattern. These differences across categories reflect that the layer-wise graphs represent a progressive refinement of functional dependencies.
> > >
> > >
> > > > **The skeleton in Figure 3 does not look like a bed skeleton.**
> > >
> > > In CAPE, the goal is not to reconstruct an object’s geometric skeleton, but to enable few-shot pose transfer from support to query. Consequently, a “structural prior” refers to task-dependent keypoint dependencies, which points are informative for each other under a given support–query pair rather than an object-dependent geometric topology. This definition follows naturally from the CAPE problem[1] formulation itself.
> > >
> > > > **Is the learned prior merely helping convergence rather than representing structure?**
> > >
> > > We clarify that the learned prior not merely serves as an optimization aid.
> > > First, ablating the structural prior while keeping the architecture and training recipe fixed leads to clear performance drops: (1) replacing i-SVAE with self-attention based adjacency reduces PCK@0.2 from 92.05 to 89.33, and (2) using a single non-iterative adjacency instead of layer-wise graphs further degrades performance (91.48 vs. 92.05). These ablations show that the model indeed relies on the learned structure to propagate structural dependencies.
> > > If the prior only helped convergence, these variants—sharing the same depth, parameter count, and training schedule would exhibit similar accuracy, which is not the case. Instead, the learned structure brings more useful information. Second, as shown in the analyses of the two categories in Figure 9, the layer-wise adjacency matrices follows a similar progression within the same category. This consistency cannot be explained by optimization alone: if the prior merely facilitated convergence, the resulting graphs would fluctuate across layers or resemble random attention patterns.
> > >
> > > **Reference**
> > >
> > > [1] Lumin Xu, Sheng Jin, Wang Zeng, et al. Pose for everything: Towards category-agnostic pose estimation. In European Conference on Computer Vision, pp. 398-416, 2022.

---

### Official Review · Reviewer_Dn8o · 2025-10-26

**Soundness:** 2
**Presentation:** 2
**Contribution:** 3
**Rating:** 8
**Confidence:** 4

**Summary:**

The submission focused on the task of category-agnostic pose estimation with few annotated example images. Specifically, the authors propose a novel generative-based framework named GenCape to estimate keypoints without additional textual descriptions or predefined skeletons. The authors propose an Structure-aware Variational Autoencoder to infer instance-specific adjacency matrices from support features, and also propose a Graph Transformer Decoder to progressively refine the estimated results. The experiments are conducted on a large-scale benchmark datasets, indicating the effectiveness of the proposed novel framework.

**Strengths:**

1. The task of category-agnostic pose estimation is interesting and fundmental for extending the category number of pose estimation.

2. The idea of using generative framework is reasonable and makes sense.

3. The proposed Structure-aware Variational Autoencoder and Compositional Graph Transfer are novel and effective to model the pose structure information.

4. The performances of proposed framework are shown on large-scale benchmark dataset, and outperform SOTA dramatically.

5. The experimental analyses are comprehensive and clear.

**Weaknesses:**

1. As discussed in Line 69-74, the support images may contain severe occlusions or incomplete annotations, and how does the proposed method address this issue? E.g., the query image has 2 occluded keypoints, while the support image has another 3 occluded keypoints. Can the proposed method estimate all the visible keypoints?

2. What's the complexity of proposed framework? It seems to be about O(M^2). Is the proposed method cost-effective?

3. Can the proposed method produce diverse results based on VAE sampling? How to understand the contradiction of diversity v.s. consistency in the proposed VAE-based method?

4. Closely related works are missing in the related works.&#x20;

   > 1. @inproceedings{chen2025weakshot, title={Weak-shot Keypoint Estimation via Keyness and Correspondence Transfer}, author={Chen, Junjie and Luo, Zeyu and Liu, Zezheng and Jiang, Wenhui and Li, Niu and Fang, Yuming}, booktitle={The Thirty-ninth Annual Conference on Neural Information Processing Systems}, year={2025} }
   >
   > 2. @inproceedings{lu2024openkd,   title={Openkd: Opening prompt diversity for zero-and few-shot keypoint detection},   author={Lu, Changsheng and Liu, Zheyuan and Koniusz, Piotr},   booktitle={European Conference on Computer Vision},  year={2024} }

**Questions:**

See Weakness.

---

> ### Author Response · Authors · 2025-11-21
>
> We sincerely thank the reviewer for the positive assessment, the encouraging score, and especially the recognition of our work’s novelty. We also appreciate the reviewer’s insightful questions, which help us clarify key aspects and further strengthen the paper. Below we provide point-by-point responses to each identified weakness.
>
> **Weakness 1**. Thank you for pointing out this important challenge. Our method is explicitly designed to handle support-query mismatch problem, including the case where the occluded keypoints in the support and query images do not overlap. The i-SVAE learns a **distribution** over latent graphs, causing occluded or unreliable support keypoints to naturally yield higher posterior variance and thus weaker adverse structural influence. The CGT then **suppresses these high-uncertainty samples** and accentuates **query-visible evidence**, ensuring the final structure aligns with the query even under mismatched occlusions.
>
> For example, in Figure 4 (row 1), the support is occluded at the bison head while the query is occluded at the hind leg. Our GenCape correctly localizes all visible keypoints, whereas GraphCape misses several predictions under this mismatch. This illustrates that our generative, uncertainty-aware design provides strong robustness when the occluded keypoints in support and query differ.
>
> **Weakness 2**. We appreciate the reviewer’s attention to computational efficiency. Although the adjacency matrix is $M\times M$, the additional cost introduced by our framework is negligible. The proposed i-SVAE encoder and decoder are lightweight MLPs (Table 8), giving a complexity of $O(MD + D_zM + M^2)$, where the quadratic term remains small for $M=100$.
> To further clarify computational efficiency, we provide a quantitative comparison of GFLOPs, parameter counts, and inference speed:
>
> **Table R1**: Comparison of efficiency and accuracy. We test the efficiency under 1-shot setting.
> | **Method**              | **GFLOPs** | **Params** | **FPS** | **PCK** |
> |------------------------|-----------:|-----------:|--------:| --------:|
> | POMNet                 | 38.01      | 48.21 M     | 6.80   | 46.05 |
> | One-Stage              | 22.65      | 26.86 M    | 36.90   | - |
> | CapeFormer             | 23.68      | 31.14 M    | 26.09   | 89.45 |
> | GraphCape-T         | 15.48      | 43.68 M    | 15.36   | 91.19 |
> | GraphCape-S       | 27.75      | 65.06 M    | 10.45   | 94.73 |
> | **GenCape-T**  | 15.66      | 44.47 M    | 14.89   | 92.05 |
> | **GenCape-S** | 27.93      | 65.85 M    | 10.44   | 95.23 |
>
> Table R1 shows that **GenCape introduces virtually no extra computational burden** compared with the representative baseline GraphCape.
>
> **Weakness 3**. Yes, our method can produce diverse structural hypotheses, because the i-SVAE samples multiple latent graphs from the posterior distribution $q_\phi(z \mid F_s) = \mathcal{N}(\mu, \mathrm{diag}(\sigma^2))$. Diversity is controlled by the non-zero posterior variance $\sigma^2$, while consistency is enforced by KL regularization towards an isotropic Guassian and sparsity on the decoded adjacency.
> To quantitatively verify this diversity–consistency trade-off, we use pretrained GenCape-T model and randomly select 275 testing samples, computing their corresponding latent embeddings, and measuring pairwise Euclidean distances, we find that the **mean distance is 1.019** (indicating samples do not collapse → **diversity**) and the **variance is 0.186** (indicating the samples form a tight cluster → **consistency**). <!--This results indicate that different samples are not collapsing to a single point (non-zero mean distance → *diversity*), yet they lie in a relatively compact region of the latent space (moderate variance → *high cohesion*).-->
>
> **Weakness 4**. We thank the reviewer for bringing these relevant works to our attention. We have included both WeakShot (Chen et al., NeurIPS 2025) and OpenKD (Lu et al., ECCV 2024) in the Related Work section in the revised manuscript.

---

> > ### Comment · Reviewer_Dn8o · 2025-11-26
> >
> > Most of my concerns are well addressed.  After reading all the comments and rebuttals, I have no extra concerns.
> >
> > For Weakness 3., only the analysis on latent embeddings is provided, but the analysis on final performance is more straightforward and important. For example, single sampling may result in sub-optimal performance. Multiple sampling may increase the computation complex. This is a particular issue of the proposed VAE-based method, and thus should be more deeply discussed.

---

> > > ### Author Response · Authors · 2025-11-30
> > >
> > > Thank you for the suggestion. In addition to Table 5, we have further included experiments with $N_s=1$ and reported the corresponding computational complexity, as shown in Table R9. Since the latent dimension is small ($D_z=32$), increasing the number of samples does not introduce parameter growth, and the additional computational overhead is negligible. However, increasing the sampling turn from 1 to 3 yields nearly a 1% performance improvement.
> > >
> > > **Table R9**: Complexity analysis under different sampling turns.
> > > | $N_s$ | 1      | 2      | 3      | 4      | 5      |
> > > |-------------------|--------|--------|--------|--------|--------|
> > > | **PCK**           | 91.09  | 91.60  | 92.05  | 91.47  | 91.63  |
> > > | **Params**        | 44.47 M| 44.47 M| 44.47 M| 44.47 M| 44.47 M|
> > > | **GFLOPs**        | 15.62  | 15.64  | 15.66  | 15.68  | 15.70  |

---

### Author Response · Authors · 2025-12-03
**Summary of Revisions in Manuscript**

We thank the reviewers for their time and the thoughtful, constructive feedback on our manuscript. Following the first revision, we have incorporated additional content to the revised manuscript to address further questions raised during the rebuttal. We believe these updates further strengthen the quality and clarity of our work. Below we summarise all changes we made compared to the original manuscript. **The newly added and revised content (compared to the original manuscript) is marked in red.**

### Content Revisions
- Added **missing reference of OpenKD and Weak-Shot** in Section 2. (addressing reviewer Dn8o)
- Added **analysis on computational complexity** in Appendix B.4 (addressing reviewer Dn8o, cNF7)
- Enhanced **the clarification of how our method addresses the query–support mismatch problem** in Section 3. (addressing reviewer Dn8o, 5bdU)
- Revised **the interpretation of Figure 3 and Figure 9** in Section 4.4 and Appendix C. (addressing reviewer EZ4n, )
- Revised **the typos and equation 4,6,7** in Section 3. (addressing reviewer EZ4n, 5bdU)
- Added **the clarification of $P_q^{(l)}$** in the Section 3.2. (addressing reviewer 5bdU)
- Added **figure illustration of CGT module** in Figure 5. (addressing reviewer 5bdU)
- Added **failure cases analyzes** in Appendix C. (addressing reviewer cNF7)

### Additional Experimental Results
- Added **non-iterative graph and self-attention as adjacency ablation studies** in Table 9. (addressing reviewer EZ4n, 5bdU)
- Added **standard cross-supercategory protocol experiments and cross-domain experiments** in Table 11 and Table 12, respectively. (addressing reviewer cNF7)
- Added **more pose estimation metrics** comparisons between GraphCape and GenCape in Table 13. (addressing reviewer cNF7)

### Additional Visualization
- Revised **skeleton visualization and qualitative results** in Figure 3. (addressing reviewer EZ4n, 5bdU)
- Added **more skelleton visualizations** in Figure 9. (addressing reviewer EZ4n, 5bdU)

We hope these revisions address all concerns and further strengthen the paper.

---

### Author Response · Authors · 2025-12-03
**Summary of Review and Rebuttal (1/2)**

Dear PCs, SACs, ACs and Reviewers,

Thank you very much for your valuable contributions. To assist the new AC in the decision-making process, we summarize here the status of our rebuttal prior to the interruption of the discussion phase. We outline the key strengths and weaknesses identified by the reviewers, and details how each concern was addressed in our responses. The full, reviewer-specific responses are available in the corresponding discussion threads.

---

## Overall Status before Rollback

Overall, we are grateful for the broadly positive assessment of our work (as average rating is 6), and recognized the value of our structure generative framework. Specifically:

- **Novelty and Conceptual Contribution.** The proposed i-SVAE and CGT are novel and effective mechanisms for modeling pose structure (**Dn8o**: Strength 3; **5bdU**: Strength 2). They consider the structure generative framework provide interesting and practically relevant insights beyond existing CAPE methods. (**Dn8o**: Strength 1,2; **E4Zn**: Strength 1; **cNF7**: Strengths)

- **Clarity of Writing and Empirical Analysis.** All reviewers who commented on presentation praised the paper as clearly written and easy to follow, with a solid and practical solution (**E4Zn**: Strength 2, **5bdU**: Strength 1). The experimental section was described as comprehensive and clear, and the key design choices were regarded as well justified through ablation studies (Tables 4–6) (**Dn8o**: Strength 5; **5bdU**: Strength 4), reinforcing the technical soundness of the work.

- **Solid Performance.** Reviewers consistently acknowledged that our method achieves state-of-the-art performance on the large-scale MP-100 benchmark. (**Dn8o**: Strength 4; **E4Zn**: Strength 3; **5bdU**: Strength 3; **cNF7**: Strengths)

After our rebuttal and the inclusion of extensive additional experiments, two Reviewers (Dn8o, cNF7) explicitly stated that their concerns had been largely resolved.
The remaining Reviewers (E4Zn, 5bdU) were unable to continue the discussion due to the premature termination of the rebuttal period caused by the information-leak incident.
Notably, both Reviewer E4Zn and 5bdU raised additional questions based on the initial discussion and indicated that **they would raise their scores upon seeing the concerns addressed**, but the process was halted before they could provide updated assessments.
As a result, **the scores remained the same as their initial values: 4 (E4Zn), 8 (Dn8o), 6 (5bdU), 6 (cNF7).**

---

## Weaknesses / Questions and Our Addressing
### 1. Analysis of computational complexity
> The reviewers requested additional comparisons on computational efficiency, such as inference time, FLOPs, model size, or throughput against baselines. (**Dn8o**: Weakness 2; **cNF7**: Weakness 2)

**Our response**: We provided the comparison of GFLOPs, parameter counts, and inference speed in Table R1 in the rebuttal. These results show that GenCape achieves a strong accuracy-efficiency trade-off: it delivers notable PCK improvements over GraphCape while keeping GFLOPs, parameter counts, and FPS essentially unchanged.

---

> ### Author Response · Authors · 2025-12-03
> **Summary of Review and Rebuttal (2/2)**
>
> ### 2. Request for more experimental results
> > The reviewers suggested additional experiments to better validate the generalization and further strength the paper.
>
> **Our response**:
>
> - **Self-attention vs i-SVAE.** To explain why self-attention can't learn the relevant connections between keypoints, we performed an additional experiment where the attention weights in self attention were used as the adjacency. (**E4Zn**: Weakness 1)
> - **Diverse VAE sampling.** We used pretrained GenCape-T model and randomly select 275 testing samples, computing their corresponding latent embeddings, and measuring pairwise Euclidean distances to quantitatively verify this diversity–consistency trade-off. (**Dn8o**: Weakness 3; **E4Zn**: Question 1)
> - **Non-iter graph prediction.** We conducted the requested ablation by replacing our layer-wise i-SVAE updates with the first-layer adjacency matrix predicted only once from the encoder output. (**E4Zn**: Weakness 2)
> - **Standard Super-category protocol.** Following the standard super-category partitioning protocol, we conducted the experiments in Table R2&R3.
> Our method achieves the best performance across all splits, demonstrating its **strong generalization**. (**cNF7**: Weakness 1)
> - **Analysis on failure cases.** We find that the primary cause of the localization errors is visual feature ambiguity, while structural inference failures play a secondary role. We took the challenging swivelchair category, and occlusion scenario as typical failure examples. We compared the GraphCape and our method on swivelchair. And we also directly tested trained GenCape-S model under random query-image masking. (**cNF7**: Question 1,2)
> - **Different query transformations.** We conducted controlled tests with scaling and rotation applied to the query images (Table R7), showing that GenCape remains **highly robust** under both transformations. (**cNF7**: Question 3)
> - **Additional metrics.** We further evaluate our model using three standard keypoint localization metrics (AUC, EPE, NME), summarized in Table R8.(**cNF7**: Question 4)
>
> ### 3. Adjacency matrix visualization hard to understand
> > The reviewers pointed out that adjacency-matrix visualizations alone are hard to interpret and requested overlaying the skeletons on the images for clearer qualitative insight. In the second-round rebuttal, the reviewer further recommended using thicker lines to indicate higher-weight connections.
>
> **Our response**:
> We have revised the paper. Following the advice, we have redrawn the skeleton visualizations in Figure 3 and Figure 9, where thicker connection indicate larger weights, and any weight below 0.5 is treated as no connection. Meanwhile, we have also added corresponding analysis. (**E4Zn**: Weakness 3; **5bdU**: Weakness 2)
>
> ### 4. Typos
> > The reviewers carefully pointed out several typos.
>
> **Our response**:
> We appreciate the reviewers for the careful effort and apologize for these oversights. We have double-checked and corrected all typos, including those in Equations 1, 4, 6 and Figures 4 and 6. (**EZ4n**: Weakness 4; **5bdU**: Weakness 1)
>
> ### 5. Missing figure illustration of CGT
> > The reviewer suggested adding CGT to Figure 2, noting that showing only $A^{(l)}_{\text{fused}}$ is not enough to easily follow.
>
> **Our response**:
> We have added a visual illustration of CGT in the Figure 5 in Appendix A.1 to further clarify its design and intuition. (**5bdU**: Weakness 1)
>
> ---
>
> ## Recognition of the Revision
> Following our responses and revisions, below are the reviewers’ responses:
>
> Reviewer Dn8o on Nov 26: "**Most of my concerns are well addressed.** ..., I have no extra concerns".
>
> Reviewer E4Zn on Nov 23: "I thank the authors for their detailed comments and added experiments. While **most of my initial concerns were addressed**, I still have a major concern remaining...". However, due to the unexpected interruption of the rebuttal, my response on Nov 25 did not receive a follow-up reply.
>
> Reviewer 5bdU on Nov 26: "Thank you for your detailed response ... This visualization would better illustrate the semantic relationships captured by these matrices". We deeply regret that the discussion was unable to proceed further because of the unexpected interruption.
>
> Reviewer cNF7 on Nov 27: "Thank you for the rebuttal and the additional experiments. **Most of my concerns have been adequately addressed,** ... provide strong support for the method’s effectiveness and generalization capability".
>
> We hope this summary assists the AC in their final assessment. We once again thank the reviewers for their constructive feedback, which has substantially strengthened our paper.

---

### Meta-Review · Area_Chair_bEwE · 2026-01-06

**Summary:**

This paper studies the category-agnostic pose estimation problem. To solve this problem, the authors propose a generative-based framework that infers keypoint relationships solely from image-based support inputs, without additional textual descriptions or predefined skeletons. Reviewers Dn8o and cNF7's concerns have been addressed during the rebuttal period. Most of Reviewer 5bdU's concerns were addressed. However, the “good vs. bad adjacency” interpretation remains qualitative. The authors should strengthen this with clearer evaluations and additional illustrative examples. Reviewer E4Zn's concern, whether the learned adjacency is truly a structural prior if it varies across layers and differs across instances that appear structurally similar, has not been fully addressed. The authors need to provide further results to explain that. During the initial reviewing stage, this paper received overall positive ratings. The only negative rating is from Reviewer E4Zn. I think they're able to raise their rating if the authors are able to address Reviewer E4Zn's concerns in the final version. I think the overall quality of this paper is impressive. Therefore, I recommend accepting this paper. The authors need to include the discussions and additional results in the final version.

**Reviewer Concerns:**

Reviewers Dn8o and cNF7's concerns have been addressed during the rebuttal period. Most of Reviewer 5bdU's concerns were addressed. However, the “good vs. bad adjacency” interpretation remains qualitative. The authors should strengthen this with clearer evaluations and additional illustrative examples. Reviewer E4Zn's concern, whether the learned adjacency is truly a structural prior if it varies across layers and differs across instances that appear structurally similar, has not been fully addressed. The authors need to provide further results to explain that.

**Reviewer Scores:**

During the initial reviewing stage, this paper received overall positive ratings. The only negative rating is from Reviewer E4Zn. I think they're able to raise their rating if the authors are able to address Reviewer E4Zn's concerns in the final version.

---

### Decision · Program_Chairs · 2026-01-26

Accept (Poster)